psychology

strabismus, three-dimensional vision, depth perception, pictorial perception, perspective cue

**Author for correspondence:**
Giedre Zlatkute
e-mail: gz37@st-andrews.ac.uk

# Unimpaired perception of relative depth from perspective cues in strabismus

Giedre Zlatkute[1], Vanessa Charlotte Sagnay de la Bastida[2] and Dhanraj Vishwanath[1]

[1]School of Psychology and Neuroscience, University of St Andrews, St Mary's Quad, St Andrews, Fife KY16 9JP, UK
[2]Institute of Psychiatry, Psychology and Neuroscience, King's College London, 16 De Crespigny Park, London SE5 8AF, UK

GZ, 0000-0002-9263-5681

Strabismus is a relatively common ophthalmological condition where the coordination of eye muscles to binocularly fixate a single point in space is impaired. This leads to deficits in vision and particularly in three-dimensional (3D) space perception. The exact nature of the deficits in 3D perception is poorly understood as much of understanding has relied on anecdotal reports or conjecture. Here, we investigated, for the first time, the perception of relative depth comparing strabismic and typically developed binocular observers. Specifically, we assessed the susceptibility to the depth cue of perspective convergence as well as the capacity to use this cue to make accurate judgements of relative depth. Susceptibility was measured by examining a 3D bias in making two-dimensional (2D) interval equidistance judgements and accuracy was measured by examining 3D interval equidistance judgements. We tested both monocular and binocular viewing of images of perspective scenes under two different psychophysical methods: two-alternative forced-choice (2AFC) and the method of adjustment. The biasing effect of perspective information on the 2D judgements (3D cue susceptibility) was highly significant and comparable for both subject groups in both the psychophysical tasks (all $p$s < 0.001) with no statistically significant difference found between the two groups. Both groups showed an underestimation in the 3D task with no significant difference between the group's judgements in the 2AFC task, but a small statistically significant difference (ratio difference of approx. 10%, $p = 0.016$) in the method of adjustment task. A small but significant effect of viewing condition (monocular versus binocular) was revealed only in the non-strabismic group (ratio difference of approx. 6%,

$p = 0.002$). Our results show that both the automatic susceptibility to, and accuracy in the use of, the perspective convergence cue in strabismus is largely comparable to that found in typically developed binocular vision, and have implications on the nature of the encoding of depth in the human visual system.

# 1. Introduction

Strabismus (a.k.a. squint) is an ophthalmological condition of the misalignment of the eyes. Due to a range of neurological or physiological issues ranging from extra-ocular muscle paralysis to developmental delays and also genetic conditions such as Down's syndrome, individuals with strabismus are unable to coordinate their eye muscles to fixate the two eyes on a single target point in space [1]. This results in the two eyes receiving information from two non-corresponding areas of space which can lead to difficulties in vision and particularly three-dimensional (3D) space perception [2]. Understanding the extent of perceptual and functional deficits arising from this binocular deficit is important considering that close to 5% of the population are diagnosed with strabismus [3,4].

Strabismus disrupts sensory fusion, the cortical process of combining the images from the two eyes into a single binocular image [3–6]. The main perceptual consequences of lack of fused binocular images is diplopia (double vision) and a lack of binocular depth perception. If strabismus is congenital or manifests in early childhood, the human visual system finds ways of adapting and avoiding the ambiguity of double vision. In most cases, this leads to the cortical suppression of one eye's image resulting in amblyopia, while under more specific circumstances, such as infantile constant strabismus, anomalous retinal correspondence or 'new' fovea may develop [2,4]. The lack of binocular fusion also implies that binocular disparities cannot be used to compute depth and induce the characteristic impression of stereopsis. While there is extensive literature on the consequences and treatment of amblyopia [7–13], the assessment of deficits in depth perception in strabismus has been more limited. Until recently, much of the understanding of the qualitative and quantitative perception of depth in individuals with strabismus has relied on anecdotal reports and conjecture.

Popular media accounts [14,15] combined with the limited empirical studies, paint a mixed picture regarding the specific capacities of strabismics to perceive depth. Some conjectures in popular media and textbooks, along with early studies, suggest that despite the absence of binocular depth perception, strabismics can perceive depth on the basis of monocular cues [5,14,16,17]. Some recent studies suggest that despite the absence of measurable stereoacuity, observers are even able to appreciate the '3D effect' associated with stereopsis in video images that contain depth information from motion [18–20]. By contrast, the most detailed and widely known subjective report of depth perception of an individual with infantile strabismus suggests that perceived depth in strabismus from monocular cues is significantly inferior to depth perception derived from binocular disparities at the qualitative, quantitative and functional level [15]. Barry, a neuroscientist, had no stereovision for most of her life but recovered normal binocular correspondence and stereopsis in late adulthood (age 47) through a series of orthoptic exercises. Barry's introspective reports comparing her perception prior to acquiring stereopsis and after suggest that the depth perceived even from monocular cues (e.g. when watching a two-dimensional (2D) movie) is inferior in strabismus and probably inferred in a top-down manner [15].

## 1.1. Varieties of depth perception

In order to better understand the nature of depth perception capacities and deficits in strabismus, it is first necessary to discriminate between the different varieties of depth perception that an individual may experience. Depth perception is often described in textbooks as the ability to judge the distance of objects and the spatial relationship among objects at different distances [21]. The initial part of the definition refers to the perception of egocentric distance (or simply distance perception), namely the distance between an observer and the viewed objects. The second part of the definition refers to depth perception, i.e. the perception of separations in depth between one object and another or between different parts of a single object. A further distinction can be made between relative and absolute (scaled) depth perception [22,23]. Relative depth is the capacity to perceive the depth relations among points in space without knowing the actual values of depth separation. Relative depth can further be broken down into the perception of depth order (ordinal depth) which is simply perceiving the ordering of objects in depth without any impression of the degree of separation

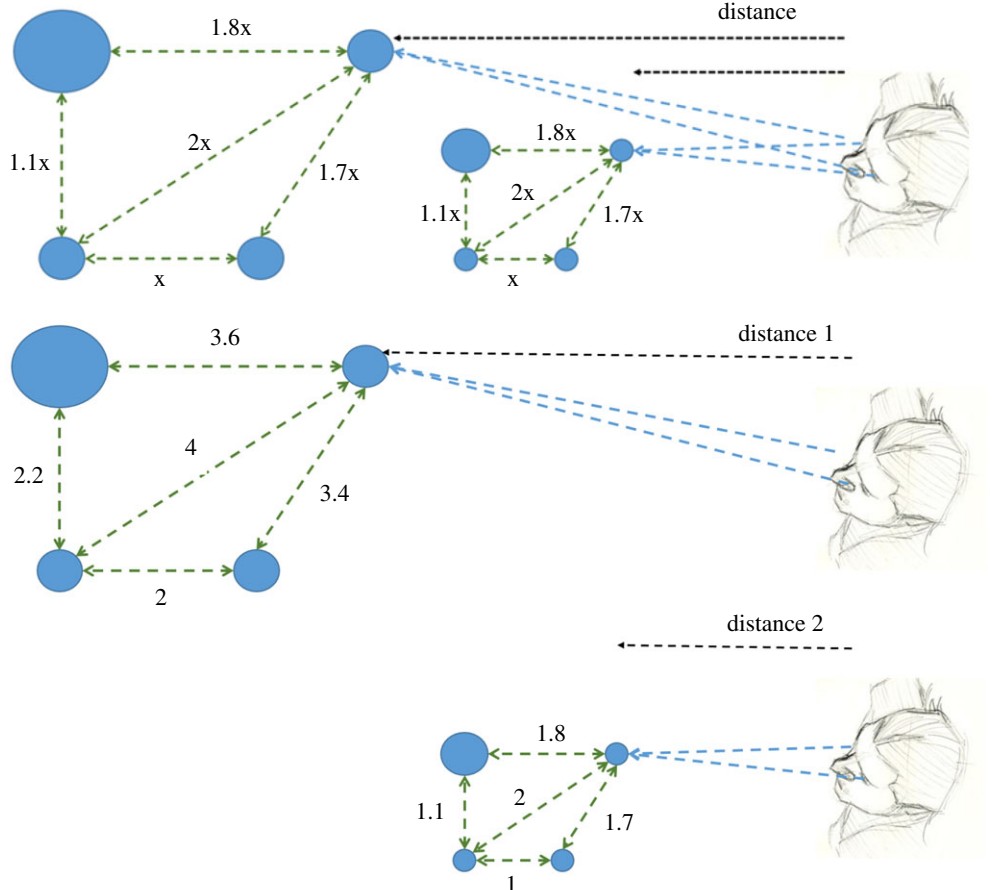

**Figure 1.** The distinction between the perception of relative and scaled (absolute) depth. In the top panel, the observer has information about the relative ratios of separations between the four objects, but not the actual values of the separations (units of distance). The observer's perception of 3D shape and layout is, therefore, ambiguous up to a uniform scaling factor. In the two lower panels, the observer perceives the actual separations between the objects in some egocentric units (units of distance) relevant for interaction.

between them, and the perception of depth ratios (perceiving the relative magnitudes of separations in depth within or among objects). The perception of 3D object shape or layout constitutes the perception of relative depth ratios. Absolute (scaled) depth is the perception of the actual depth separations between objects or points within an object scaled to some egocentric motor metric and is required in order to execute visuomotor tasks such as grasping (see figure 1).

Egocentric distance perception in near space (less than 2 m) is believed to rely primarily on extra-retinal cues such as accommodation, binocular convergence and defocus blur, while the judgement of the egocentric distance of objects located on the ground plane in farther space (5–25 m) appears to rely on ground plane information and declination from eye level [24–31]. Since convergence is impaired in strabismus, judging distances in near space should be compromised, while the perception of far distances along the ground plane is not expected to be compromised since it is based primarily on visual declination. Recent studies that have examined distance perception in strabismics for near and far space are consistent with this [28,32,33].

Within near (personal) space, perception of egocentric distance has been assessed by having subjects perform high-precision visuomotor tasks without any haptic feedback [32–34]. Consistent with the lack of binocular convergence information, clear deficits are seen in high-precision tasks within reach space where subjects misjudge the distance to slots when placing pegs and misjudge the distance of a bead to be threaded [32,33]. For far distances, strabismics show the normal ability to blind walk to a previewed target placed on the ground but show deficits in performing these tasks when the same target object is suspended mid-air, a condition which putatively complicates the derivation of distance purely from the ground plane declination information [28].

The predictions with respect to relative and scaled (absolute) depth, however, are more complex. Both monocular depth cues (motion parallax, perspective convergence, texture gradients, relative size,

shading) and binocular disparities specify only relative depth relations and need to be scaled by egocentric distance cues such as convergence in order to estimate scaled (absolute) depth.

Due to strabismics' lack of normal binocular convergence, both the distance cue of vergence and the depth cue of binocular disparity cannot be used, which should compromise the perception of absolute depth at near distances affecting tasks such as grasping or tasks that require nulling of absolute depth such as threading a needle. Consistent with this, deficits in bead threading (requiring absolute depth nulling) have been demonstrated in strabismics [32,33]. However, to our knowledge, no direct examination of absolute depth perception in strabismics has been conducted by measuring either pantomimed grip apertures or online measurement of grasping dynamics.

Despite deficits in the use of the depth cue of binocular disparity, strabismics should, in principle, be able to use pictorial monocular depth cues (perspective, shading, texture) and motion parallax cues to make judgements of relative depth relations such as 3D shape, surface slant and curvature in depth. However, detailed anecdotal reports suggest that relative depth perception from monocular cues (e.g. motion parallax, shading, perspective) is compromised in strabismus [15]. Specifically, these descriptions suggest that stereotypical observers are able to use monocular cues more effectively (for example when watching standard 2D movies) because they can use 'a lifetime of past visual experiences (with binocular vision) to re-create the missing stereo information' (Barry [15, p. 102]). These reports support a more general view that only binocular disparity 'directly' provides a bottom-up quantitative perception of depth, while depth from monocular (pictorial) cues rely on indirect top-down inferences [35], putatively built up through experience with 3D perception from binocular disparity. If the latter were true, we would expect strabismic observer to be impaired in making such top-down inferences of depth using monocular cues due to their lack of learned correlation of the cues with depth from disparity. Moreover, there is significant evidence that different forms of 3D perception (distance, relative depth and absolute depth) might be dissociated and/or processed separately in the visual pathways [22,23,26,27,36] with potentially distinct developmental trajectories. Strabismus may affect each of them in different ways under different conditions. To our knowledge, no studies have examined the perception of relative depth based on monocular cues in strabismus.

In this study, we aimed to examine, for the first time, the use of a monocular depth cue for the judgement of relative depth in strabismus. Specifically, we aimed to determine the bottom-up susceptibility to the monocular depth cue of perspective convergence as well as the capacity to use the cue to make accurate judgements of relative depth in pictorial images comparing strabismic observers to those with normal stereovision. While pictorial images are routinely used to examine the role of monocular cues in depth perception, it is generally believed that binocular viewing of pictorial images by observers with normal stereovision will yield shallower estimates of depth than what the cue specifies because of the conflicting disparity cue specifying the flat picture surface. This belief suggests that strabismics who cannot use binocular disparity might perceive more depth in pictorial images than those with normal stereovision. However, most empirical evidence shows no difference in judgements of depth comparing monocular and binocular viewing for a range of tasks ([37–40]; though see Koenderink *et al.* [41]). Thus, if strabismics have the same capacity to use relative depth cues, we hypothesize no difference in judgements between the two groups under both monocular and binocular viewing.

## 1.2. Susceptibility to monocular depth cues and the capacity to use them for accurate judgements

The consensus view of monocular (pictorial) depth cues in observers with normal stereovision is that they provide bottom-up quantitative visual information for depth perception, though there are alternative views [35]. A persuasive argument for the consensus view is that when viewing pictorial images that contain monocular depth cues, depth is perceived despite the strong conflicting information from binocular disparity signalling a flat surface [42,43], experimental settings where disparity can be overridden by monocular depth cues [44], and the fact that these cues combine in a lawful way with binocular disparity information [45]. The automatic susceptibility to monocular depth cues is further highlighted in observers' inability to cognitively suppress perspective cues in making judgements of object size and shape in pictorial images. Observers show systematic misperception of 2D object size and shape, with their judgements falling between the actual 2D stimulation and the implied 3D configuration, despite being instructed to make judgements on only the 2D shape. Thouless [46] referred to this automatic tendency as 'phenomenal regression to the real object' whose magnitude is measured by the Thouless ratio. Such an automatic susceptibility to

(a)    (b)

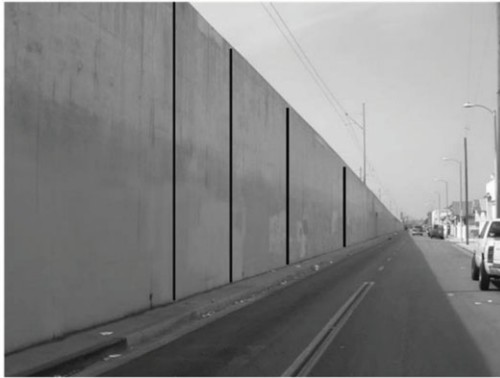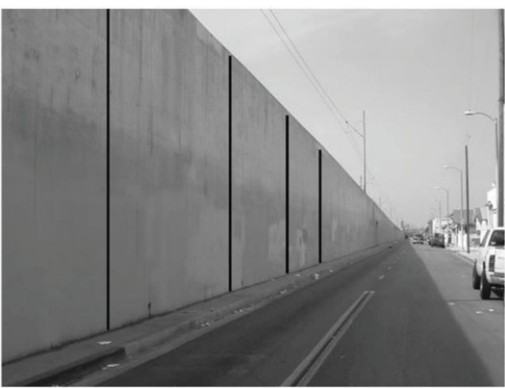

**Figure 2.** Demonstration of the Thouless-type regression to the real object when judging interval equidistance of lines superimposed on a perspective pictorial image. The required equidistance judgement is to determine if the separations between the four lines on the 2D plane of the page (ignoring the 3D pictorial content) are equal. (a) The intervals are equidistant on the 2D plane of the surface of the photograph but are typically perceived to be non-equidistant. (b) The spacing consistent with correct equidistant spacing in 3D pictorial space. Images adapted from Fig. 5 in Erkelens [47].

monocular depth cue of perspective convergence is demonstrated in figure 2. Measuring the Thouless ratio in such stimulus conditions can, therefore, help determine if strabismics are subject to the same bottom-up susceptibility to monocular relative depth cues as stereotypical observers or if they rely on cognitive inferences which can easily be suppressed through cognitive effort.

Another way to identify differences between strabismics and non-strabismics is to measure the capacity to make accurate relative depth judgements derived from monocular cues. Judgements of relative depth on the basis of monocular cues typically show underestimation in comparison to ground truth even in stereotypical observers [48], and specifically for perspective convergence [37,47,49]. Determining if strabismics show the same or greater level of underestimation will help identify processing differences between the two groups.

Both these measures (susceptibility and accuracy), taken together, can provide evidence of differences in the early and/or late processing of monocular depth cues between strabismic and stereotypical observers. In the present study, we report on two experiments where we examine differences in the susceptibility to, and accuracy in the use of, one of the primary monocular cues to relative depth perception: perspective convergence (linear perspective).

# 2. Experiment 1

In this experiment, we compared strabismic and non-strabismic participants' susceptibility to a 3D bias in making 2D interval equidistance judgements in the presence of perspective convergence cues as well as their capacity to use perspective information to make accurate judgements of interval equidistance in a depicted 3D scene.

## 2.1. Methods

### 2.1.1. Participant recruitment

The current research was performed without access to participants' medical records, thus the only information available during recruitment about their prior visual condition and diagnosis was provided by participants themselves. Both strabismus and amblyopia are umbrella terms for a number of conditions whose types and causes can significantly vary between individuals. Amblyopia onset is most often associated with strabismus in early childhood and persists into adulthood, as adult amblyopes could have both constant as well as intermitted strabismus. Thus, there is a close link and considerable overlap between the two conditions [4,11,50]. The initial recruitment of participants was done by asking for (i) participants who have no issues using 3D technology (e.g. watching 3D movies) and who have no known history of strabismus or amblyopia; (ii) participants who self-reported current or past diagnosis of strabismus, or have experienced difficulty or inability to use 3D technology (including double vision).

Participant eligibility to take part in the experiment and further classification was determined using binocular stereovision tests. Additional information was gathered about participants' experiences when trying to view 3D movies, or any daily difficulties with their vision, including regular experiences of double vision or eye misalignment when feeling tired or sick. Due to high variance in manifestations of strabismus and the function-focused approach taken by orthoptists, intermittent strabismus remains undiagnosed if it does not affect patients' daily activities [3,5,51,52]. Therefore, individuals who had been diagnosed only with amblyopia, but self-reported regular experiences of double vision and/or notable eye misalignment (e.g. observable in photographs, or noted by family members, friends), were included in the strabismic participant group if they also showed no or limited binocular stereoacuity based on the screening testing carried out for these experiments.

In summary, the classification of participants into the strabismic group was based on self-reported regular experiences of double vision, history with strabismus and/or amblyopia, as well as severely impaired or no clinically measurable binocular stereoacuity (more detail on classification according to visual screening tests is given in §2.1.2). We, therefore, use the classification terms 'strabismic participants' and 'participants with no/limited stereovision' interchangeably.

### 2.1.2. Participants and their classification

In total, 32 individuals took part in this experiment. Twenty-four normal stereovision subjects with typically developed binocular vision. Eight were individuals with no or limited stereovision, one of whom was an author of the current paper.

Individuals with strabismus can be classified into different groups either based on the history of strabismus and the angle of deviation, or according to the level of measurable binocular stereovision. Due to the high variability among participants' visual history and lack of clinical records, the classification for the experiment was based on the level of participants' binocular stereovision. Information about participants' vision was gathered using a number of ophthalmological tests as well as self-reports about participants' clinical history, and self-reported daily experiences, such as double vision.

The visual acuity was measured at 10 feet distance using the Snellen chart. The Western Ophthalmics near distance visual acuity test was used at approximately 40 cm to measure near distance visual acuity (should see line 15 J1), and at 60 cm to measure visual acuity at the monitor viewing distance (should see line 20–25 J2–J3). Participants wore their corresponding habitual corrections during the testing, for instance, reading glasses for the near distance visual acuity test. The aim of these tests was to ensure that participants are able to see the provided visual stimuli at least with their dominant eye. Eye dominance was identified using the Miles test.

Literature discussing clinical binocular stereovision tests highlights their limitations and the fact that, currently, there is no single perfect test for measuring binocular stereovision [18,53,54]. Therefore, three stereotests were used to measure binocular stereovision: Titmus (Fly), TNO and RanDot. These tests are widely used and measure both types of binocular stereovision: global and local [53,54]. The RanDot stereotest was used to measure stereoacuity of all participants, while individuals with strabismus were also tested using the Fly (a.k.a. Titmus) and TNO stereotests, thus assessing the full extent of stereovision deficits [53]. The type of strabismus was determined using cover–uncover tests as well as self-reported history of the disorder.

The first three stereoscopic circles in the Titmus test (800 and 200 arcsec) have clear monocular feature changes and thus successful identification is not considered to be diagnostic of the presence of binocular stereoacuity [4,5]. Therefore, participants were classified as having binocular stereovision if they had no measurable stereovision in RanDot and TNO tests, and could correctly identify none or only the first three stereoscopic circles in the Titmus test. Similarly, the initial stimuli on the RanDot and TNO scale have large binocular disparities, which may be processed monocularly if participants alternate their eyes, move their head or the test sheet [4,5,51]. Participants were, therefore, classified as having limited binocular vision if they responded correctly to no more than four first stimuli on the RanDot scale (up to 400 arcsec), and no more than the first four of the TNO test (first pairs, up to 240 arcsec) consistent with prior literature that suggest that success in identifying the initial levels cannot necessarily be associated with binocular stereovision [4,5,18,53,54]. Participants who did not exceed these levels of binocular stereovision according the global stereovision tests, and who responded to no more than half of the local stereovision test stimuli (Titmus test less than 100 arcsec) were classified as having limited binocular stereovision.

control control perspective pictorial perspective

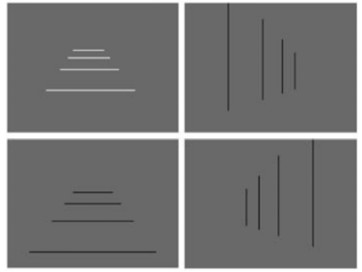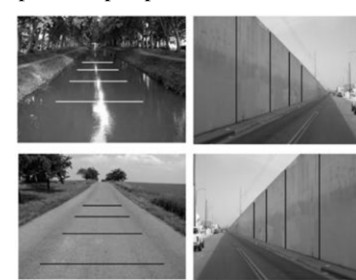

**Figure 3.** The three stimulus conditions and their variants (control, control perspective and pictorial perspective) used in Experiment 1.

### 2.1.3. Stimuli

Stimuli created for the current experiment were similar to the photographic images of real scenes used by Erkelens [47] to examine the perception of depth and distance from perspective cues in pictorial space. Photographic images of both horizontal and vertical surfaces in depth were tested (horizontal surface: road or canal, vertical surfaces: rectilinear wall). The horizontal surface was tested in two orientations (receding leftward or rightward) by simply mirror reversing the original image. Four black or white lines were superimposed on the photographic image to create the probes to measure perceived spatial interval equidistance. These images of the photograph superimposed with the probe lines are referred to as the pictorial perspective (PP) condition. Two additional control stimuli images were created (figure 3). In the control (C) condition, the probes were four lines equal in length, displayed either horizontally or vertically. In the control perspective (CP) condition, the probes were tapering lines matching exactly the length of the probe lines in the PP condition but without the background image. The colours of the probe lines were matched to those of the PP condition for each of the respective C conditions. In total, the C condition had two unique stimuli, while the CP and PP had four unique stimuli each (figure 3).

### 2.1.4. The interval ratio

For each of the unique stimuli, 32 images were created with differing ratios of separation between the left-most and right-most spatial intervals formed by the four-probe lines. For example, for the vertically oriented wall surface (tilt = 0), the ratio range was 0.2497 (Image 1) to 1.2234 (image 32). Ratio 1.00 (image 29) was veridical judgement for interval equidistance on the 2D plane, and ratio 0.317 (image 4) was the veridical judgement for interval equidistance in the depicted 3D space. Examples of these ratios are shown using lines overlaid on a photographic perspective image of a wall (figure 4).

 The dependent measure for the perceived interval equidistance was the ratio of the judged interval and the comparison interval (figure 5). The middle interval always stayed the same. The total ratio range was from 0.25 (Image 1) to 1.22 (image 32). Ratio 1.00 (image 29) was veridical judgement for interval equidistance on the 2D plane, and ratio 0.32 (image 4) was veridical judgement for interval equidistance in the depicted 3D space.

### 2.1.5. Stimulus presentation

Stimuli were presented on a 1920 × 1080 px resolution LCD monitor (size of monitor, and size of image on monitor) set at a fixed distance of 60 cm. Participants viewed images binocularly under normal lighting conditions, with their head stabilized on a chin/head rest. The entire monitor, including its frame, was visible during the task performance.

### 2.1.6. Task

Participants were asked to make interval equidistance judgements between lines either on the 2D plane of the monitor or in the depicted 3D pictorial space. In the first part of the experiment, participants were asked to make judgements on the 2D plane of the monitor ignoring pictorial information in the picture as best as they could. Using a two-alternative forced-choice (2AFC) staircase method, participants judged

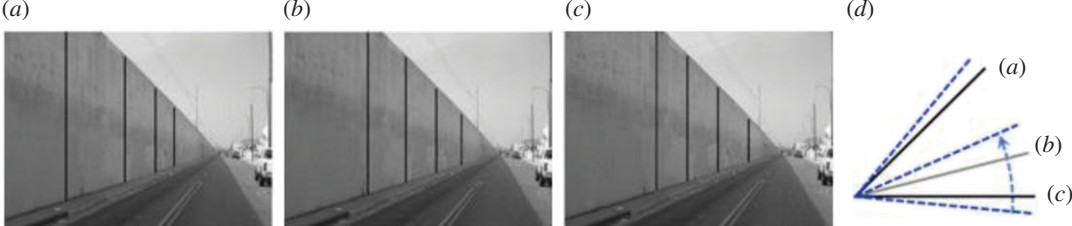

**Figure 4.** Example of pictorial perspective images with different intervals between the left and right pair of lines. The interval between the middle two lines remains constant across the images. (*a*) Line separations consistent with equidistance on the depicted surface in 3D scene (derived using a standard method using vanishing points [55]). (*b*) Line separations that are equidistant for a depicted surface slant that is less than slant depicted in the 3D scene (underestimated slant). (*c*) Shows lines that are equidistant in the depicted 3D space. (*d*) Diagrammatic representation of the derivation of the line intervals assuming different slants ranging from slightly less than slant 0 (equidistance on the plane of the picture (*c*)) to slightly greater than the depicted 3D slant (*a*). Stimuli were constructed for a total of 32 steps.

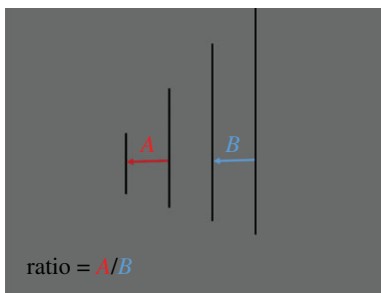

**Figure 5.** The dependent measure: the interval equidistance ratio: *A*/*B*, where *A* is the judged interval, and *B* is the comparison interval. (Example stimuli from 2D CP condition).

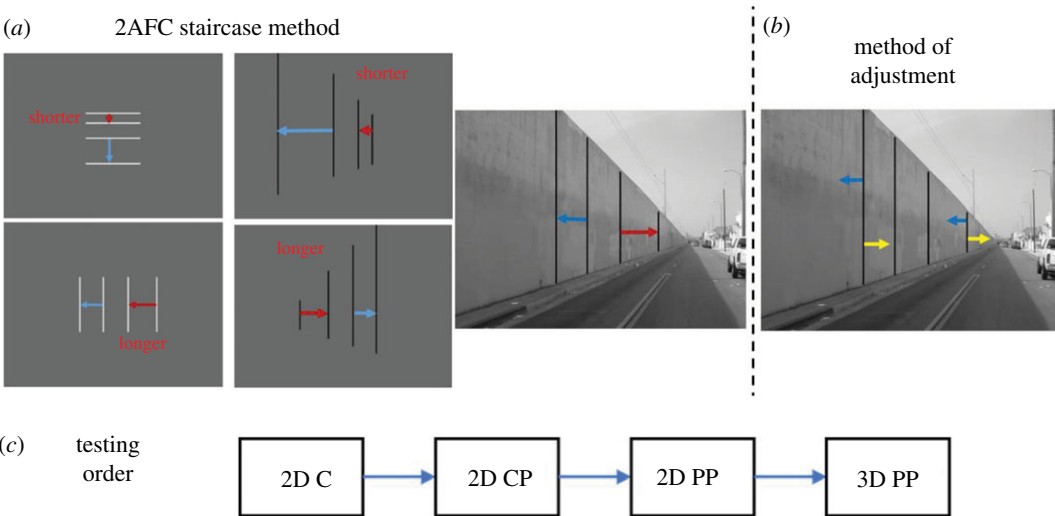

**Figure 6.** Summary of tasks and testing order in Experiment 1. (*a*) Examples for judgements to be made in 2D control, 2D control perspective and 2D pictorial perspective conditions two-alternative forced-choice method tasks. (*b*) Examples of judgements to be made in 3D pictorial perspective condition using a method of adjustment. (*c*) Indicates the testing order.

one of the intervals as being shorter or longer than the second interval (figure 6*a*). In the second part of the experiment, participants were asked to adjust the position of the lines (method of adjustment) until the intervals appeared equidistant within the depicted 3D space, by using the mouse keys to move the outer flanking lines (figure 6*b*).

### 2.1.7. Procedure

Participants were given instructions using a PowerPoint presentation before each task. Participants were asked to make 2D interval equidistance judgements using the staircase procedure in the following order: control (2D C), control perspective (2D CP) and pictorial perspective (2D PP) conditions (figure 6c). Before each of these conditions, instructions specified which interval participants were to judge against which reference interval (examples in figure 6a). Participants were instructed to ignore the background image in the 2D PP condition and perform the task judging, as best they could, the actual physical interval spacing of the lines on the 2D plane of the monitor. Participants initiated the trial via a keyboard key press, at which time the stimulus was displayed for a total of 1.75s. When the stimulus disappeared, participants responded via mouse keys (right key for longer, left for shorter). Participants were free to make eye movements. One second after their response, a new stimulus would appear. There were two staircases for each unique stimulus (e.g. left versus right oriented), terminating after five reversals. The presentation of different stimulus orientations was randomly interleaved.

For the 3D pictorial perspective (3D PP) judgement task participants were instructed to adjust the lines so that they appeared to be equally separated within the 3D space depicted in the pictorial image. Participants adjusted the positions of the two outer flanking lines by pressing the two mouse keys and submitted their response via a keyboard key press. The stimulus display time and number of position adjustments were unlimited. Each of the PP stimulus images (four) was presented four times each, in random order, for a total of 16 adjustment trials.

In all blocks, participants viewed the stimuli binocularly and were instructed that they were not required to maintain fixation on any specific location but were permitted to look around the image, if needed, while the stimulus was displayed, in order to make their judgement.

### 2.1.8. Stimuli tested

In the 2D judgement tasks using the two-alternative forced-choice staircase method, there were two one-up one-down staircases for each unique stimulus, terminating at the maximum count of five reversals. Staircase step size was 3 units (on the 32 step range) until the second reversal after which step size reduced to 1 unit. The last three reversal points were averaged to determine the judged setting for the statistical analyses. In the 3D method-of-adjustment task, each of the four stimuli was presented four times; thus, in total containing 16 adjustment trials. Each key press adjustment resulted in a change of size of 1 unit. Stimuli in all the tasks were presented in a computer-generated randomized sequence.

## 2.2. Results

### 2.2.1. Participants

Three normal stereovision participants' data were excluded from the analysis due to the high variability in responses compared with the other subjects (greater than 0.3 difference in ratios for the same stimuli), which indicated that they were probably not following instructions or attending to the task (further discussed in limitations). Another normal stereovision participant was eliminated due to an incomplete dataset. Therefore, 20 normal stereovision participants (average age in years 21, s.d. 2.91) were used for the analysis. All normal stereovision participants responded correctly to all RanDot test stimuli (20 arcsec). The last nine subjects that were recruited were also tested on the two additional binocular stereoacuity tests, and responded correctly to all Titmus test stimuli (40 arcsec) and to the majority of the TNO stimuli (on average 60 arcsec), matching the clinical criteria for having normal binocular stereovision [53,54].

From the total eight participants with no/limited binocular stereovision (average age in years 34, s.d. 13.28) an author of the paper was eliminated as a non-naive subject. More information about strabismic participants with no/limited binocular stereovision is provided in table 1.

### 2.2.2. Interval equidistance on a two-dimensional plane

Figure 7 summarizes the average ratio thresholds at which intervals were perceived to be equidistant for the two groups: typical binocular vision and those with a history of strabismus with no or limited stereovision. All participants were able to perform the task accurately in the 2D C condition and showed ratio thresholds close to the veridical ratio of one (average = 1.02, s.d. = 0.047). For the

**Table 1.** Summary information of participants with no/limited binocular stereovision tested in Experiment 1.

| no. | age (years) | disorder history (self-reported) | onset | deviation at time of testing | corrections (self-reported) | visual acuity far | near | monitor | eye dominance | stereovision Titmus (Fly) | TNO | RanDot |
|---|---|---|---|---|---|---|---|---|---|---|---|---|
| 1.[a,b] | 27 | L amblyopia far-sighted | infantile | none | patching 6/7 to 14 years | R: 20/10 L: 20/10 | R: 15J1 L: 15J1 | R: 20J2 L: 20J2 | right | 200″ | none | none |
| 2.[a] | 48 | amblyopia R eye | not known | exophoria | patching, corrective lens | R: 20/15 L: 20/13 | R: 60J10 L: 20J1 | R: 60J10 L: 25J3 | left | 200″ | none | none |
| 3.[a] | 42 | strabismus R eye | infantile | L eye hypertropia | surgeries at 1–2 and 11–12 years; patching, drops | R: 20/15 L: 20/13 | R: 20J2 L: 20J2 | R: 20J2 L: 20J2 | right | none | none | none |
| 4.[a] | 54 | strabismus R amblyopia | infantile | R eye esotropia | surgery at 4 years, early patching | R: none L: 20/13 | R: none L: 20J2 | R: none L: 25J3 | left | none | none | none |
| 5.[a] | 43 | strabismus R eye | infantile | R eye esotropia | surgery at around 1 year; early patching | R: 20/15 L: 20/10 | R: 50J8 L:15J1 | R: 50J8 L: 20J2 | left | none | none | none |
| 6. | 20 | esotropia L eye | infantile | L eye esohypotropia | surgery at around 2 years, patching prior it | R: 20/10 L: 20/15 | R: 15J1 L: 20J2 | R: 20J2 L: 25J3 | right | none | none | none |
| 7.[c] | 18 | short-sighted | N/A | none | none | R: 20/15 L: 20/15 | R: 30J4 L: 25J3 | R: 40J7 L: 30J4 | left | 400″ | 240″ | 100″ |
| 8.[a,c] | 21 | R amblyopia short-sighted | not known | none | patching 4/5 to 9/10 years | R: 20/20 L: 20/13 | R: 40J7 L: 15J1 | R: 50J8 L: 20J2 | left | 140″ | 480″ | none |

[a]Tested in both Experiments 1 and 2.

[b]Non-naïve subject eliminated from analysis.

[c]Limited binocular stereovision.

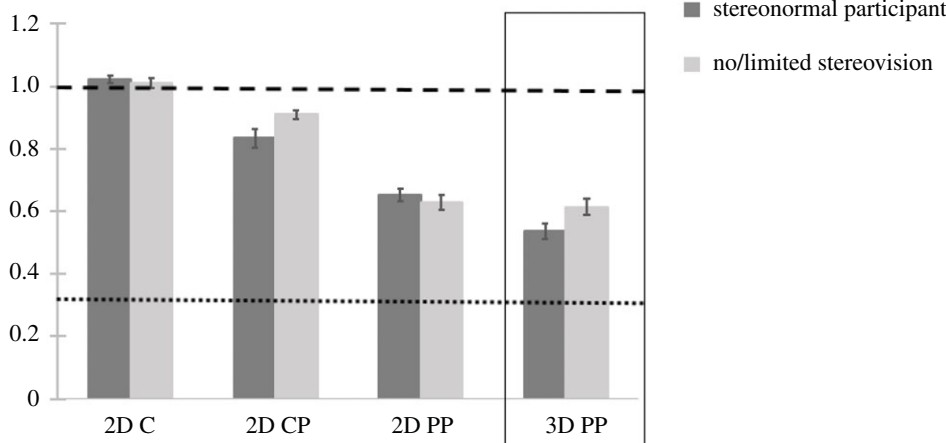

**Figure 7.** Average ratios at which intervals were perceived to be equidistant on a 2D plane of the monitor (2D conditions), and in the depicted 3D pictorial space (3D condition). Error bars indicate standard error of the mean. Dashed line indicates veridical ratio for 2D judgements, while dotted line indicates veridical 3D judgements.

ANOVA analysis Mauchly's test of sphericity was non-significant ($p = 0.076$). Results from a $2 \times 3$ mixed design ANOVA (between subjects factor: subject type (non-strabismic = 20, strabismic = 7); within subjects factor: perspective condition (2D C, 2D CP, 2D PP)) showed that judgements about interval equidistance on a 2D plane of the monitor were affected by perspective information in the stimuli $F_{2,50} = 107.104$, $p < 0.001$. Levene's test of equality of variances across subject groups was significant only for the 2D CP condition ($p = 0.004$) and non-significant for the 2D C and 2D PP conditions (Levene's test $p = 0.710$ and $p = 0.251$, respectively).

Pairwise comparisons between stimulus conditions (with Bonferroni corrections) showed that differences between all pairs of conditions were significant (2D C versus 2D CP, mean difference = 0.145, $p < 0.001$; 2D C versus 2D PP, mean difference = 0.377, $p < 0.001$; 2D CP versus 2D PP, mean difference = 0.232, $p < 0.001$). There was no main effect of subject type $F_{1,25} = 0.397$, $p > 0.5$ and no interaction between the two factors $F_{2,50} = 2.244$, $p > 0.1$.

For the separate repeated measures ANOVA analysis Mauchly's tests of sphericity were non-significant for both subject groups (stereonormal subjects $p = 0.110$; no/limited stereovision subjects $p = 0.418$). ANOVA run for each subject group revealed that the effect of perspective information was significant for both groups taken separately: non-strabismic subjects $F_{2,38} = 78.18$, $p < 0.001$, and strabismic subjects with no/limited stereovision $F_{2,12} = 210.86$, $p < 0.001$.

The sparse perspective convergence information in 2D CP condition had a small but significant effect (approx. 15% error) in both groups (planned pairwise comparisons with Bonferroni corrections, $p \leq 0.001$). The 2D PP condition had an unexpectedly large effect, resulting in nearly a 40% error on average in the interval ratio. In both groups (analysed together and separately), the 2D PP ratio was significantly smaller than in both the 2D CP condition and the 2D C condition, and the 2D CP ratio was significantly smaller than for the 2D C condition (Bonferroni-corrected pairwise comparisons, all $p$s < 0.001).

### 2.2.3. Interval equidistance in three-dimensional pictorial space

The participants' ratios for judgements in a 3D pictorial space (3D PP) were similar in value to 2D PP judgements (figure 7). For individuals with no/limited stereovision, the perceived amount of pictorial depth (3D PP ratio average = 0.61, s.d. = 0.066) was almost identical to the interval equidistance error in 2D PP (ratio average = 0.63, s.d. = 0.065). For normal stereovision subjects, this difference was larger: 3D PP ratio average = 0.54, s.d. = 0.103; 2D PP ratio average = 0.65, s.d. = 0.089. However, an independent sample $t$-test showed that difference between the normal stereovision and no/limited stereovision participants' depth judgements (3D PP) was not significant ($t_{25} = 1.051$, $p = 0.083$). Since the 3D PP condition used the adjustment method, it could not be directly compared with 2D conditions, which were measured using the staircase method. This is further discussed in the limitations below and addressed in Experiment 2.

## 2.3. Discussion and limitations

Participants accurately judged interval equidistance on the 2D plane of the monitor when there was no perspective information in the stimuli. However, their responses were significantly affected by perspective convergence information, even when they were required to ignore it. Moreover, sparse and decontextualized perspective convergence cue information was sufficient to significantly affect participants' 2D interval equidistance judgements. All individuals, despite the varying levels of binocular stereoacuity were susceptible to perspective convergence information, even when required to ignore it. This suggests that perspective convergence cues entail an automatic bottom-up processing and in the case of typical stereovision observers, the resulting bias in 2D judgements cannot be suppressed even when highly reliable binocular disparity information could have contributed to a less biased judgement. There were no significant differences between individuals with strabismus and typically developed binocular individuals when asked to make judgements about interval equidistance either in 2D space or within the depicted 3D pictorial space, suggesting that strabismics do not have any impairments with regards to the use of perspective cues for judging depth. However, the difference in the size of the subject pool for the two groups, and further limitations discussed below, suggest caution in interpretation.

First, participants' ratios for the 3D interval judgements were highly similar to their 2D interval equidistance judgements (2D PP vs 3D PP). It is plausible that this similarity is due to a methodological issue, rather than actual lack of perceptual differentiation between the 2D and 3D judgements. Specifically, it is possible that the order of testing, the 2D task followed by the 3D task, may have caused participants to misinterpret the initial 2D task to be a 3D task because of the strong pictorial background (in comparison to the control and CP condition where the 2D instruction was more salient). This may explain the relatively large bias in the 2D PP condition which yielded ratios almost the same as the 3D PP judgement. A reversed order in which subjects do the more natural 3D task first and are then asked to do a different 2D task ignoring the pictorial information may have made the latter instruction more salient. Secondly, due to the fact that this task was done under binocular viewing, the negligible difference between both the 2D bias and 3D accuracy between strabismics and stereotypical observers may not necessarily reflect equivalent susceptibility to and capacity for the use of perspective convergence information. The possibility remains that the effectiveness of the perspective convergence cue is indeed higher in stereotypical observers, but its effect in both the 2D and 3D judgements was moderated by the conflicting disparity cue that signals a flat surface. The presence of conflicting depth information, based on the conventional understanding of depth cue integration [56], is that perceived depth from perspective cues in pictures for stereotypical observers under binocular viewing would be suppressed. While recent evidence indicates that this is not the case, since judgements under monocular and binocular viewing of depth from monocular cues has not been found to be measurably different [37–40], it is still plausible that there could have been larger differences in both susceptibility and accuracy between the two groups, but that this difference was masked because conflicting disparity information is present in one but not the other group. It is, therefore, important to compare susceptibility to the perspective cue (2D judgement bias) and its effectiveness (accuracy of 3D judgements) for both monocular and binocular viewing in both these subject groups. Finally, the 2D PP and 3D PP tasks used different psychophysical methods (method of limits and method of adjustment, respectively) and the exposure durations for the two tasks was different. In Experiment 2, we addressed these issues by altering task order to eliminate any potential misunderstanding of the 2D versus 3D task, testing both monocular and binocular viewing, and using both the method of limits and method of adjustment.

# 3. Experiment 2

## 3.1. Methods

### 3.1.1. Stimuli and their presentation

All display and stimulus conditions were the same as the previous experiment except as described. All tasks within this experiment were performed under two viewing conditions: binocular and monocular. In the monocular condition, subjects wore an eye patch over their non-dominant eye. The order of testing (monocular versus binocular) was randomized between participants.

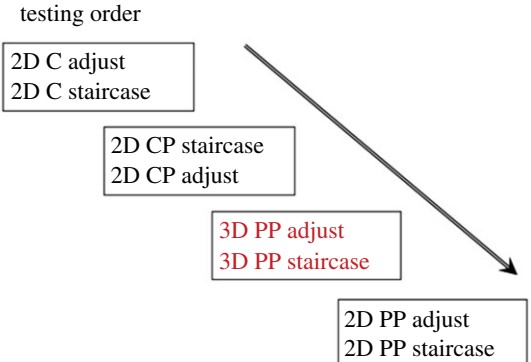

testing order

**Figure 8.** Testing order for Experiment 2.

### 3.1.2. Task and procedure

As in Experiment 1, participants were asked to judge the equidistance of intervals either on the 2D plane of the monitor or the depicted 3D space. We used the method of adjustment (MoA) with unlimited stimulus exposure times for the 3D task, reasoning that it would be difficult to make the more complex 3D judgement with the short exposure of the 2AFC task for naive observers. However, in the second experiment, we decided to use both methods for all conditions, since each method may yield a different pattern of results and lead to inferences on differences between fast automatic bottom-up (2AFC) or deliberate top-down processing (MoA).

In the 2AFC staircase method, participants had to judge one interval as being either shorter or longer than the comparison interval in the 2D plane (conditions: 2D C, 2D CP, 2D PP), or in the depicted 3D space (3D PP condition). In the method of adjustment tasks, participants had to adjust the position of the lines until the intervals appeared equidistant in the 2D plane (conditions 2D C, 2D CP, 2D PP), or in the depicted 3D space of the image (condition 3D PP).

The specific testing order (figure 8) was chosen considering methodological limitations raised in Experiment 1. To allow participants to familiarize themselves with the stimuli, the adjustment tasks were performed first for all conditions with the exception of 2D CP. For the 2D CP condition, the staircase method task was performed first, as we noted that dynamic adjustment induced a sensation of three-dimensionality that is not present in the static (staircase) method, and we worried that this may cause a bias in the staircase condition if the adjustment condition was done first. Additionally, the testing order was changed for the 2D PP and 3D PP conditions, where the stimuli are identical, but subjects are asked to perform different tasks: judging when the intervals appear equidistant on the 2D plane of the monitor, and within the depicted 3D scene, respectively. We reasoned that it was better to have subjects do the 3D PP task first so that they understand what is meant by equidistance in 3D space first and can then consequently understand how the 2D task (ignore the 3D information) is different, even though the stimulus in both tasks is identical.

The current experiment followed the same procedure as Experiment 1 with the following changes. For the staircase method tasks in the 2D C condition, the stimulus was displayed for 1.25 s, while for all other conditions it was displayed for 1.75 s. This was done to reduce testing time, as 2D C involves a very straightforward task, which participants had no difficulty performing in Experiment 1. The fixation image was displayed until participants responded. In the method of adjustment tasks, the fixation image was displayed between the images for 1 s. The stimulus display time and number of positional adjustments were unlimited in the adjustment method. Participants were not asked to restrict eye movements.

### 3.1.3. Stimulus value manipulations

In the staircase method, for all stimulus conditions in the two-alternative forced-choice staircase method tasks, there were two interleaved one-up one-down staircases for each unique stimulus, terminating at the maximum count of four reversals. The staircase step sizes were adaptive: four units initially, two after the first reversal and one after the second reversal. The last three reversals were averaged to obtain the value of the judgement for analysis.

In the method of adjustment tasks, there were a total of 16 trials per stimulus type and viewing condition (monocular and binocular). In the 2D CP, 2D PP and 3D PP conditions, each unique stimulus (four in total for each) was presented four times, while in 2D C each unique stimulus (two

in total) was presented eight times. Each adjustment resulted in a change of size ratio of 1 step out of 32 possible steps (see Experiment 1).

In all the tasks, stimuli presentation order was a computer-generated randomized sequence.

### 3.1.4. Participants

In total 26 people took part in the second experiment. Twelve subjects with typically developed binocular vision. Fourteen were individuals with no or limited stereovision and a history of strabismus or related disorder, one of whom was an author of the current paper. All participants with typically developed binocular vision were new recruits and not tested in Experiment 1. Five of the strabismic participants had participated in Experiment 1 and an additional six new strabismic participants were recruited (see tables 1 and 2). Recruitment and screening criteria for the additional strabismic participants was done in the same way as in Experiment 1. All participants had their stereoacuity tested using all three stereotests: Fly (a.k.a. Titmus), TNO and RanDot.

## 3.2. Results

### 3.2.1. Participants

Four participants were excluded from the analysis: the author, due to potential experimenter bias; one normal stereovision participant due to an incomplete dataset; and two strabismic participants showing high variability in responses compared with the other strabismic participants (greater than 0.3 differences in ratios for the same stimuli) which suggested a change in criteria or limited engagement with the tasks. Consequently, 22 participants (11 in each group) were used for the analysis. The average age in years of participants with normal stereovision was 25 (s.d. 4.73), while the average age in years was 34 (s.d. 12.05) for participants with no/limited stereovision (for more information about participants' vision, see table 2).

For normal stereovision subjects, the average binocular stereoacuity according to the Titmus Fly test (total range 800–40 arcsec) was 40 arcsec (range 50–40 arcsec). According to the RanDot test (total range 400–20 arcsec) the average binocular stereoacuity of normal stereovision subjects was 23 arcsec (range 40–20 arcsec). According to the TNO test (total range 480–15 arcsec) the average binocular stereoacuity of normal stereovision subjects was 65 arcsec (range 120–30 arcsec).

### 3.2.2. General performance and task compliance

For the participants who did both experiments, it was possible to compare their performance in Experiments 1 and 2, using data for the corresponding tasks. This indicated that the changes in testing order and more detailed explanation of the tasks elicited the expected differentiation between 2D and 3D judgements in the PP conditions that was generally absent in Experiment 1 (see electronic supplementary material, figure S12). The comparison of participants' data show that while performance in 2D C and CP staircase method tasks was similar between Experiments 1 and 2, there were notable changes in participants' 2D and 3D interval equidistance judgements in PP stimuli tasks. This establishes that it was the changes in testing order and more explicit explanation of the tasks that crystallized the differences between the 2D PP and 3D PP conditions in Experiment 2.

Additionally, even though the chosen interval equidistance ratios varied among participants across the experimental conditions, individual participants' performance was generally consistent within the conditions (figure 9). The comparison of the two participants groups does not reveal major notable patterns, even though in some cases it seems that strabismic individuals overall showed less variability than controls.

### 3.2.3. Interval equidistance on the two-dimensional plane

Figure 10 summarizes the average ratio thresholds at which intervals were perceived to be equidistant, under binocular or monocular viewing, for the two groups: typically developed stereovision subjects, and subjects with no/limited stereovision. As in Experiment 1, all participants were able to perform the task accurately in the 2D C condition, when tested using either the 2AFC staircase method or method of adjustment, producing ratio thresholds close to the veridical value of one (staircase method: average = 1.001, standard error = 0.008; adjustment method: average = 0.98, standard error = 0.005). Overall, the findings of Experiment 2 replicated the results of Experiment 1, showing the same effect of perspective

**Table 2.** Summary information of participants with no/limited binocular stereovision tested in Experiment 2.

| nr. | age (years) | disorder history (self-reported) | onset | deviation at time of testing | corrections (self-reported) | visual acuity | | | eye dominance | stereovision | | |
|---|---|---|---|---|---|---|---|---|---|---|---|---|
| | | | | | | far | near | monitor | | Titmus (Fly) | TNO | RanDot |
| 1.[a,b] | 27 | L amblyopia far-sighted | infantile | none | patching 6/7 to 14 years | R: 20/10 L: 20/10 | R: 15J1 L: 15J1 | R: 20J2 L: 20J2 | right | 200″ | none | none |
| 2.[a,d] | 21 | R amblyopia short-sighted | not known | none | patching 4/5 to 9/10 years | R: 20/20 L: 20/13 | R: 40J7 L: 15J1 | R: 50J8 L: 20J2 | left | 140″ | 480″ | none |
| 3. | 30 | intermittent strabismus | not known | exophoria | none | R: 20/13 L: 20/13 | R: 20J2 L: 20J2 | R: 25J3 L: 20J2 | right | none | none | none |
| 4.[d] | 23 | strabismus L esotropia | infantile | esophoria | surgeries at 4 and 15 years | R: 20/10 L: 20/13 | R: 15J1 L: 15J1 | R: 20J2 L: 20J2 | right | 200″ | 480″ | 200″ |
| 5.[d] | 31 | strabismus R esotropia | 14 years | none | surgery at 21 years | R: 20/20 L: 20/13 | R: 20J2 L: 15J1 | R: 25J3 L: 20J2 | left | 140″ | 480″ | 100″ |
| 6.[a] | 48 | amblyopia R eye | not known | exophoria | patching, corrective lens | R: 20/15 L: 20/13 | R: 60J10 L: 20J1 | R: 60J10 L: 25J3 | Left | 200″ | none | none |
| 7. | 54 | strabismus R amblyopia | infantile | R eye esotropia | surgery at 4 years, early patching | R: none L: 20/13 | R: none L: 20J2 | R: none L: 25J3 | left | none | none | none |
| 8.[a] | 42 | strabismus R eye | infantile | L eye hypertropia | surgeries at 1–2 and 11–12 years; patching, drops | R: 20/15 L: 20/13 | R: 20J2 L: 20J2 | R: 20J2 L: 20J2 | right | none | none | none |
| 9.[c] | 24 | L eye hypertropia | infantile | R eye hypertropia | surgery at 2 years; patching 2–5 years | R: 20/13 L: 20/15 | R: 15J1 L: 15J1 | R: 20J2 L: 20J2 | right | 800″ | none | none |
| 10. | 20 | L and R esotropia | infantile | L eye esotropia esophoria | surgeries at 11 and 13 years; patching, drops | R: 20/10 L: 20/13 | R: 15J1 L: 20J2 | R: 20J2 L: 20J2 | right | none | none | none |

(Continued.)

**Table 2.** (*Continued.*)

| nr. | age (years) | disorder history (self-reported) | onset | deviation at time of testing | corrections (self-reported) | visual acuity far | near | monitor | eye dominance | stereovision Titmus (Fly) | TNO | RanDot |
|---|---|---|---|---|---|---|---|---|---|---|---|---|
| 11. | 45 | esophoria | 2 years | R eye esotropia esophoria | surgery at around 3.5 years, patching prior it | R: 20/30 L: 20/15 | R: no L: 40J7 | R: no L: 40J7 | left | 400″ | none | none |
| 12.[c,d] | 48 | esotropia R eye | infantile | none | surgery at 4 years; patching, motor exercises | R: 20/20 L: 20/15 | R: 20J2 L: 15J1 | R: 40J7 L: 30J4 | left | 200″ | 480″ | 480″ |
| 13.[a] | 43 | strabismus R eye | infantile | R eye esotropia | surgery at around 1 year; early patching | R: 20/15 L: 20/10 | R: 50J8 L: 15J1 | R: 50J8 L: 20J2 | left | none | none | none |
| 14. | 19 | L and R esotropia R amblyopia short-sighted | infantile | R exo-hypotropia exo-hypophoria | surgeries prior 1 year; patching and drops to 5 years | R: 20/25 L: 20/10 | R: 30J4 L: 15J1 | R: 50J8 L: 20J2 | left | 200″ | none | none |

[a]Tested in both Experiments 1 and 2.

[b]Non-naive subject eliminated from analysis.

[c]Outliers eliminated from analysis.

[d]Limited binocular stereovision.

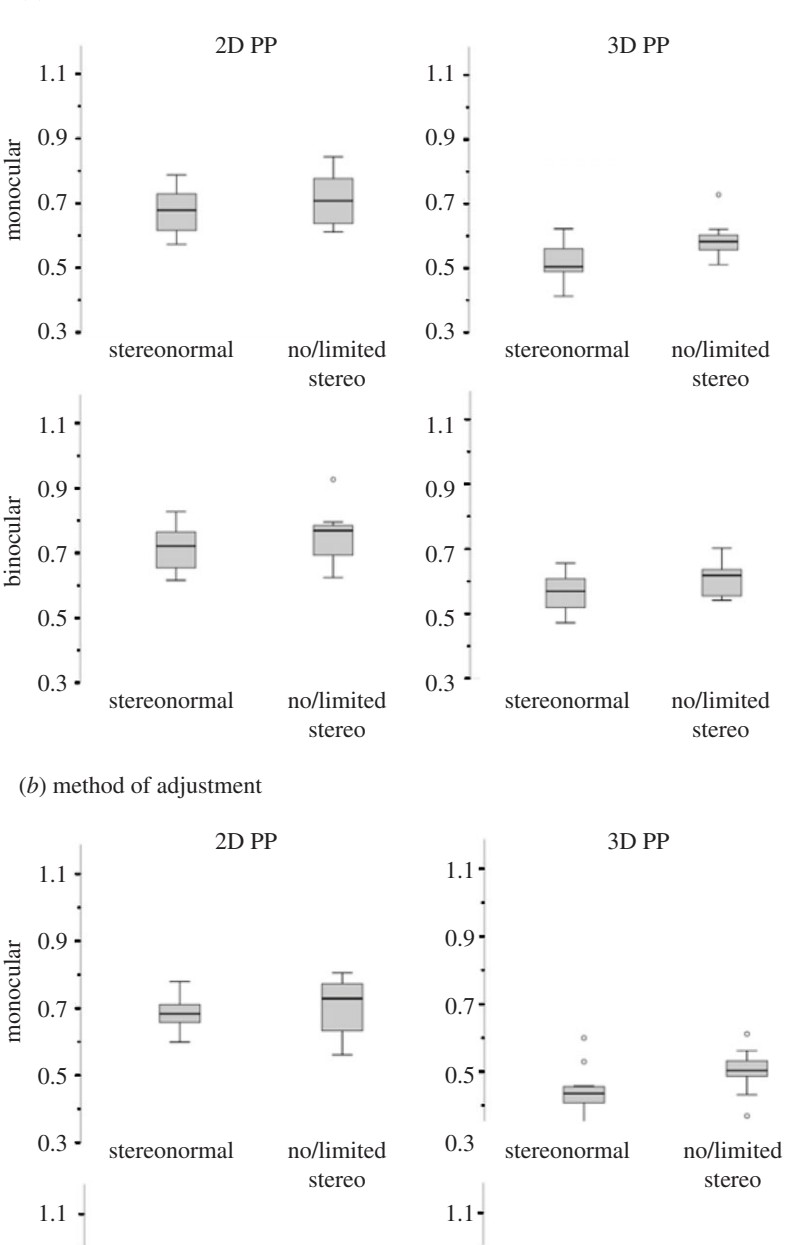

**Figure 9.** Whisker box plots summarizing the data for the two experimental conditions (2D PP, 3D PP) under monocular and binocular viewing for stereonormal subjects and subjects with no/limited stereovision. (*a*) 2AFC staircase method. (*b*) Method of adjustment. Circles mark the outliers, which are between 1.5 and 3 times the height of the box. No extreme cases (more than 3 times the height of the box) are present in the Experiment 2 data.

information on 2D equidistance judgements. It shows that this effect is generally present to the same degree regardless of viewing condition (monocular or binocular) or psychophysical procedure and exposure duration (2AFC staircase method or method of adjustment).

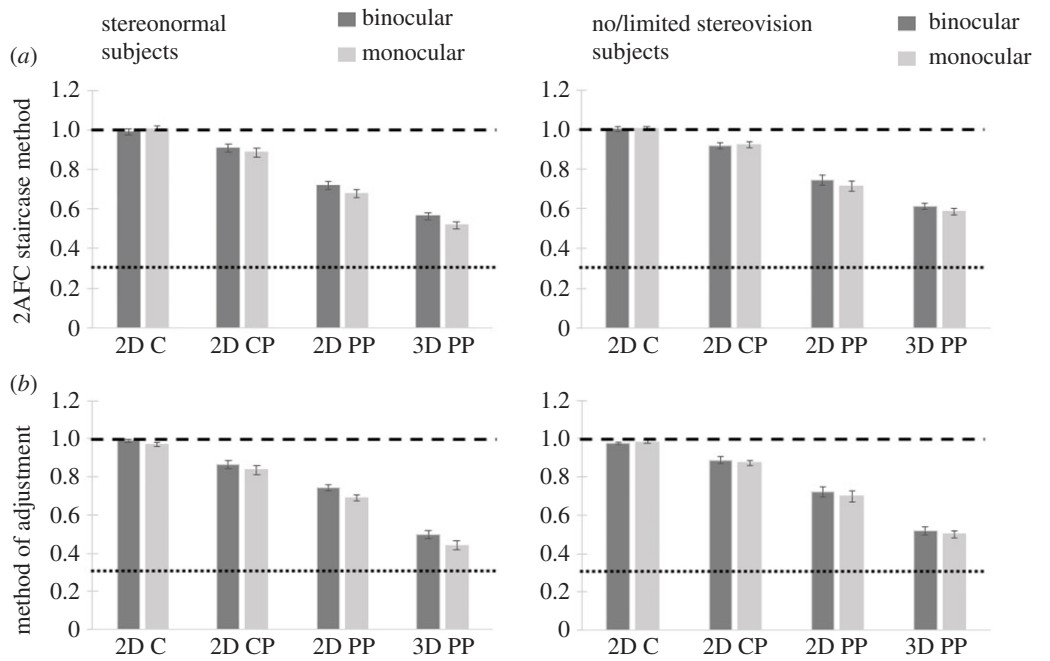

**Figure 10.** Average ratios at which intervals were perceived to be equidistant on a 2D plane of the monitor (2D conditions), and in the depicted 3D pictorial space (3D condition), for each participant group under monocular and binocular viewing. Error bars indicate standard error of the mean. Dashed line indicates veridical ratio for 2D judgements, while dotted line indicates veridical ratio for 3D judgements. (a) 2AFC staircase method. (b) Method of adjustment.

### 3.2.4. Equidistant interval judgement in two-dimensional space

For the 2AFC task ANOVA analysis, Mauchly's tests of sphericity were non-significant (perspective $p = 0.213$, perspective × viewing $p = 0.817$). The $2 \times 3 \times 2$ mixed design ANOVA with one between-subject factor (subject type: normal versus no/limited stereovision) and two within-subject factors (perspective information: C, CP, PP; viewing condition: monocular, binocular) replicated findings in Experiment 1 showing a significant main effect of perspective information ($F_{2,40} = 187.974$, $p < 0.001$), but no main effect of subject type ($F_{1,20} = 1.406$, $p = 0.250$). The interaction between factors subject type and viewing condition was non-significant ($p = 0.567$) though the main effect of viewing condition approached significance ($F_{1,20} = 4.068$, $p = 0.057$) and the interaction perspective information × viewing was significant ($F_{2,40} = 6.504$, $p = 0.004$) suggesting that there was an underlying effect of the viewing condition. Levene's tests of equality of variance were non-significant for all stimulus conditions (monocular 2D C $p = 0.310$, monocular 2D CP $p = 0.234$, monocular 2D PP $p = 0.445$, binocular 2D C $p = 0.103$, binocular 2D CP $p = 0.274$, binocular 2D PP $p = 0.615$).

For the adjustment task ANOVA analysis Mauchly's tests of sphericity were non-significant (perspective $p = 0.772$, perspective × viewing $p = 0.768$). The $2 \times 3 \times 2$ mixed design ANOVA revealed similar results, with a highly significant effect of perspective information ($F_{2,40} = 141.25$, $p < 0.001$) and a non-significant interaction between subject type and perspective information ($p = 0.51$). However, the main effect of viewing condition was significant ($F_{1,20} = 20.140$, $p < 0.001$) and the interaction perspective information × viewing was also significant ($F_{2,40} = 4.724$, $p = 0.014$) suggesting that the effect of viewing condition was stronger in the adjustment task, where subject could view the stimulus for a longer duration while making the judgement. Levene's tests of equality of error variance were non-significant for all stimulus conditions (monocular 2D C $p = 0.614$, monocular 2D CP $p = 0.083$, monocular 2D PP $p = 0.051$, binocular 2D C $p = 0.091$, binocular 2D CP $p = 0.378$, binocular 2D PP $p = 0.60$).

To examine in more detail the effect of the variables we ran separate $3 \times 2$ repeated measures ANOVAs for the two groups of subjects. Mauchly's tests of sphericity were non-significant for both testing methods for normal binocular stereovision subjects as well as participants with no/limited binocular stereovision. This analysis revealed that the effect of perspective information was less in binocular than the monocular viewing condition in the normal stereovision group and the difference was more significant in the method of adjustment task $F_{1,10} = 25.966$, $p < 0.001$) than in the 2AFC task ($F_{1,10} = 6.832$, $p = 0.026$). However, the difference between viewing conditions was non-significant for

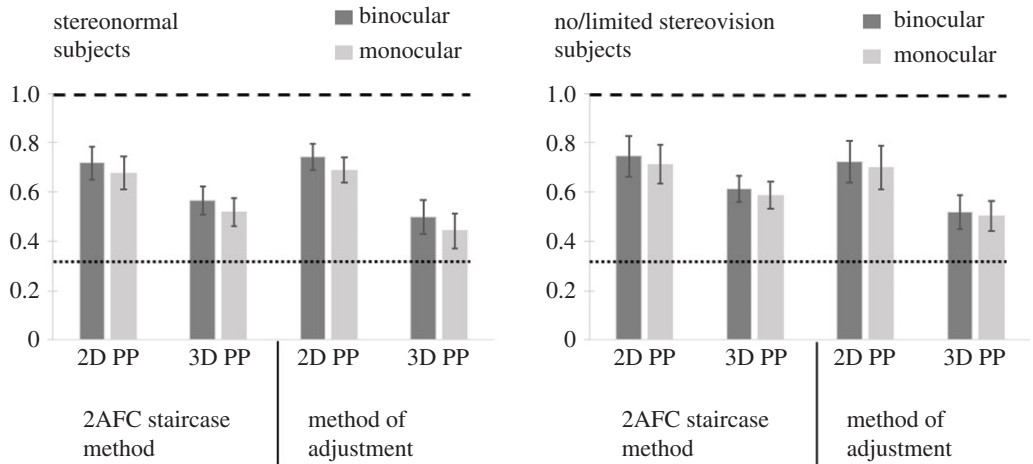

**Figure 11.** Average ratios at which intervals were perceived to be equidistant on a 2D plane of the monitor (2D PP) or in the depicted 3D pictorials space (3D PP), for 2AFC staircase and adjustment methods, under monocular and binocular viewing. Error bars indicate standard deviation. Dashed line indicates veridical ratio for 2D judgements, while dotted line indicates veridical ratio for 3D judgements.

the no/limited stereovision group in both tasks ($F_{1,10} = 0.683$, $p = 0.428$; $F_{1,10} = 1.676$, $p = 0.225$). Nevertheless, the overall effect of both sparse perspective information and PP information was strong for both groups as revealed by planned Bonferroni-corrected pairwise comparisons; all comparisons (C versus CP, C versus PP, CP versus PP) for both groups yielded $p$s < 0.001, again replicating the results from Experiment 1.

### 3.2.5. Equidistant interval judgement in three-dimensional space

For the 2AFC task a 2 × 2 mixed design ANOVA (between subject factor: group type; within subject factor: viewing condition) indicated that there was a main effect of subject group type (mean difference in ratio: 0.057; $F_{1,20} = 6.893$, $p = 0.016$) and a main effect of viewing (mean difference in ratio: 0.036; $F_{1,20} = 8.721$, $p = 0.008$) on 3D judgements. For the method of adjustment (MoA) task, there was no significant effect of subject group type (mean difference 0.04; $F_{1,20} = 1.940$, $p = 0.179$), but still there was an effect of viewing (mean difference in ratio: 0.037; $F_{1,20} = 12.763$, $p = 0.002$). There was no significant interaction between subject type and viewing condition implying that the effect of viewing condition (monocular versus binocular) was not modulated by whether the subjects had normal stereovision. Levene's tests for equality of variance were non-significant for both testing methods and for each perspective and viewing condition.

Ratios at which intervals were perceived to be equidistant in a 3D pictorial space (3D PP) were notably smaller and closer to the veridical 3D space judgements for both subject groups than any ratios chosen for 2D judgements on the same images (figure 11). In order to examine the differences between 3D PP and 2D PP equidistance judgements we ran a 2 × 2 × 2 mixed design ANOVA with one between subject factor (subject type: normal stereovision, no/limited stereo) and two within subject factors (viewing condition: monocular, binocular; judgement type: 2D, 3D). There was a significant main effect of judgement type (2D versus 3D) for both the 2AFC and MoA tasks ($F_{1,20} = 75.112$, $p < 0.001$ and $F_{1,20} = 85.036$, $p < 0.001$, respectively). There was also a main effect of viewing condition (monocular versus binocular) for both 2AFC and MoA tasks ($F_{1,20} = 19.240$, $p < 0.001$ and $F_{1,20} = 24.884$, $p < 0.001$, respectively).

Separate 2 × 2 repeated measure ANOVAs for each group (within-subject factors: viewing condition (binocular versus monocular) and judgement type (2D versus 3D)) further showed that viewing condition was a significant factor for the normal stereovision group for both the 2AFC and MoA tasks ($F_{1,10} = 23.142$, $p = 0.001$ and $F_{1,10} = 23.731$, $p = 0.001$, respectively), but not significant for the no/limited stereovision group ($F_{1,10} = 4.509$, $p = 0.06$ and $F_{1,10} = 3.820$, $p = 0.08$, respectively). The effect of judgement type (2D versus 3D) was highly significant for both subject groups for both task types (2AFC and MoA) with all $p$s < 0.001. This indicates that, in contrast to Experiment 1, and consistent with our rationale for changing the order of 2D and 3D testing, that subjects were indeed making different judgements in the two tasks.

Finally, for the 3D PP judgements, both normal stereovision and no/limited stereovision subjects appeared to be more accurate in the MoA task compared with the 2AFC task. For example, normal stereovision subjects were about 0.08 ratio units more accurate under monocular viewing and 0.07 units more accurate under binocular viewing, while no/limited stereovision subjects were more accurate by 0.08 and 0.09 units, respectively. There was no consistent effect of task type in the 2D PP judgements. This suggests that the longer viewing duration allowed for more accurate 3D interval judgements, but viewing duration did not have a consistent effect on the errors in the 2D interval judgements.

In summary, the biasing effect of perspective information on the 2D judgements (3D cue susceptibility) was highly significant and comparable for both subject groups in both of the psychophysical tasks (MoA, 2AFC) with no statistically significant difference found between the two groups. By contrast to Experiment 1, the difference between the 2D and 3D equidistance judgement (for the PP stimulus) was highly significant for both groups and tasks, indicating that the difference between the 2D and 3D judgements was effectively communicated to subjects with the change in order. The 2AFC task, but not the more accurate MoA task, showed a statistically significant difference in ratios for the 3D task between the two groups (no/limited stereovision versus normal stereovision). The viewing condition (binocular versus monocular) appeared to only significantly affect judgements for the normal stereovision group, with the biasing effect of the perspective cue in the 2D judgement being somewhat reduced in the binocular compared with the monocular viewing condition, and the 3D judgement being slightly more accurate in the monocular compared with the binocular condition. However, the ratio difference for these group-specific differences were relatively small: difference in 3D judgements between the two groups in the 2AFC task (0.057); difference in bias between monocular and binocular viewing for normal stereovision subjects in the 2D task (0.05); difference between monocular and binocular viewing for normal stereovision subjects in the 3D judgements (0.051). This contrasts to the effect sizes for the other main effects that were observed for both groups, e.g. the biasing effect of PP information on 2D judgements (2D PP versus 2D C, average difference: 0.28), the biasing effect of sparse perceptive information on 2D judgements (2D CP versus 2D C, average difference 0.10), the difference between 3D and 2D judgements (3D PP versus 2D PP, average difference: 0.22).

## 3.3. Discussion and limitations

Experiment 2 replicated the findings of Experiment 1 in relation to interval equidistance judgements on a 2D plane under two psychophysical methods. All participants were able to correctly judge interval equidistance when there was no perspective information, but with the introduction even of sparse perspective convergence information, they chose equidistance ratios significantly regressed toward the 3D interpretation. The regression toward the 3D interpretation was surprisingly large in the condition where there were rich perspective cues (PP condition). This effect was comparable for strabismics and non-strabismics.

The judgements of interval equidistance in the depicted 3D pictorial space were significantly different from equidistance judgements restricted to 2D plane (PP stimulus condition) with subjects choosing ratios closer to the veridical 3D values. This difference was significant for all participants, viewing conditions, and under both psychophysical methods, thus showing that methodological improvements in Experiment 2 have successfully differentiated the two different judgements (3D versus 2D). Therefore, the errors made in the 2D judgement tasks can more reasonably be attributed to automatic bottom-up processing of perspective information rather than lack of understanding of the judgement to be made.

For the 3D judgements, ratios for the two subject groups were comparable, but there was a small but statistically significant difference between the two groups in the 2AFC task, where normal stereovision subjects were on average more accurate by 0.057 ratio units. However, there was no statistically significant difference between the two groups in the adjustment task, in which both groups showed more accurate 3D responses. This indicates that in a task where there is unlimited time to make a judgement, performance in the two groups in 3D judgements is similar.

Only normal stereovision subjects showed a small but significant difference in ratios between monocular and binocular viewing, suggesting that the conflicting binocular disparity under binocular viewing may attenuate the automatic susceptibility to pictorial depth cues in a 2D task and that removing this conflicting information (monocular viewing) improves slightly the accuracy in 3D judgement. However, this difference (about 0.05 ratio units) was relatively small in comparison with the main effect of biasing from the perceptive cue for 2D judgements (2D PP versus 2D C: 0.28) and the shift in ratios for 3D judgements (3D PP versus 2D C: 0.5) that was observed in both groups. This

is consistent with recent evidence showing little or no effect of conflicting binocular disparity information in the perception of pictorial depth [37–40].

No systematic differences between two types of psychophysical method (2AFC and MoA) were evident in the 2D judgements, but both subject groups were slightly more accurate in making 3D judgements in the MoA compared with the 2AFC task. This suggests that while the extended duration of exposure may have helped in judging more accurately the line separations in 3D space, observers are not able to cognitively override the susceptibility to monocular cues in pictorial images in the 2D judgements, even under longer duration exposures, suggesting that the regression to the 3D interpretation cannot be suppressed, implicating bottom-up processing of the perspective cue in both groups.

# 4. General discussion

Overall, the results of the two experiments suggest that both the susceptibility to and the capacity to use perspective information is largely comparable in strabismus compared with typically developed binocular vision. Both experiments found subjects with little or no measurable stereovision to be susceptible to pictorial depth information, including even sparse perspective convergence information, to the same degree as subjects with typically developed binocularity. Similarly, strabismic observers showed similar capacity as normal stereovision observers in the use of perspective cues to make relative depth judgements in depicted 3D space, showing no statistically significant difference in the MoA task in which both groups were more accurate for the 3D judgements.

The main systematic difference between the two groups was found in Experiment 2 where there was a statistically significant interaction between the subject group and viewing condition (binocular versus monocular). Specifically, normal stereovision observers showed a larger error in 3D judgements (regression to the 2D judgement) under binocular compared with monocular viewing and a smaller bias in 2D judgements under binocular viewing. However, these differences (of the order of about 0.05 ratio units), were relatively small, compared with even the overall biasing effect of sparse decontextualized convergence information on depth judgements on 2D judgements. This indicates that despite binocular disparity information strongly favouring a 2D interpretation (in normal stereovision subjects), the automatically processed pictorial depth is not 'suppressed' to the degree that would be predicted on the basis of normative weighted cue averaging models [45,56]. A more plausible explanation is at the task rather than perceptual level. We relied on instructions to direct subjects to do either the 2D or 3D interval equidistance task, given the inherent ambiguity of the tasks in a pictorial image. The presence of binocular disparities specifying a flat (2D) picture surface for the normal stereovision subjects is likely to have had some effect in priming the 2D task interpretation despite instructions, potentially underlying the small shift of ratios in the direction of a 2D interpretation for both the 2D and 3D task. This small difference should, therefore, be interpreted with caution given that previous work has shown no influence of conflicting binocular disparity information in 3D shape judgements in pictorial images [37–40].

The use of two different testing methods did reveal, as expected, that when participants were given more time (method of adjustment) they made slightly more accurate relative depth in 3D judgements. Whether this is simply a methodological effect related to degree of attention that can be allocated to make judgements within a limited period of time or suggests a role of cognitive processing in depth judgements will require further examination. However, the fact that this difference was evident in both groups discounts an interpretation that strabismic observers specifically are more reliant on cognitive inferences (cf. Barry [15]). However, Barry's introspective reports provide powerful and important insights that must be reconciled with the empirical findings as they probably relate to complex differences in the subjective impression of depth that may be independent of the capacity to make judgements of relative depth (see below).

Taken together, our results show for the first time that susceptibly to and capacity for the use of the primary static monocular cue to depth perception (perspective convergence) is not compromised in strabismus. There are, no doubt, significant individual differences in susceptibility to and capacity in the use of 3D cues, something that has been demonstrated repeatedly even among typically developed binocular observers [57]. Such individual differences are bound to be even larger within a cohort of individuals with limited or no stereovision, given the differences in aetiology. However, considered broadly, our results do not reveal any notable differences between these two groups in relative depth judgements based on an important monocular cue.

This adds to the accumulating empirical evidence pointing to important similarities and differences in the perception of depth in strabismic observers with limited or no stereovision in comparison with those with typically developed binocular stereovision. So far, evidence suggests that the perception of distance along the ground plane (which uses the cue of declination from eye level) and the perception of relative depth from the perspective is intact in strabismus (Ooi & He [28] and current study). However, the perception of egocentric distance in near (manual interaction space) space, which putatively depends on vergence, is compromised, as demonstrated by studies requiring precision judgements of distance from the observer [32–34]. Furthermore, the perception of distance of an object in far space is compromised when the object does not contact the ground [28], suggesting that such judgements may require some form of binocular information. More generally, these results are consistent with the view that human depth perception relies on multiple distinct encodings.

Two important components of depth perception in strabismic observers, however, remain to be studied. One is the perception of scaled (absolute) depth, for example, through the measurement of pantomimed grip apertures or grasp kinematics when aiming to grasp objects in manual interaction space. The other is the nature of the qualitative experience of space and 3D objects in strabismus and to understand how this compares with the impression of depth in pictures or in the presence of stereopsis [23].

Finally, a more general interpretation of our result pertains to the question of whether monocular cues are genuine bottom-up depth cues, or instead, are developmentally reliant on cognitive inferences or learning in conjunction with disparity-specified depth. Our results which show large effects of the 'regression to the real object' [46] in 3D space in both typically developed binocular observers and strabismics, even those with no history of binocular depth perception, suggests that monocular cues such as perspective are bottom-up perceptual cues that cannot be suppressed even with considerable cognitive effort. This also bears on the debate on whether the perception of pictorial depth, in general, is genuine bottom-up depth perception is or simply some sort of cognitive inference or convention, a view that has been revived lately [35]. The fact that all observers regardless of their developmental history are unable to cognitively suppress the effect of monocular depth cues, even in the presence of conflicting binocular information, supports the long-held view that pictorial depth is indeed genuine perception [42,58,59].

Ethics. The University of St Andrews School of Psychology and Neuroscience Ethics Committee delegated to act on behalf of the University Teaching and Research Ethics Committee (UTREC) has approved the ethical application for this study. Original approval code: PS11855.

Data accessibility. Research data underpinning this work constituted part of G. Zlatkute's PhD thesis. Original codes and stimuli are available at: https://doi.org/10.17630/dd47de98-b469-42c4-b3de-d0cf733158f6 (see data for Chapter 5: relative depth perception). The information about participants due to ethical limitations is provided as an anonymous summary in the electronic supplementary material.

Authors' contributions. G.Z. contributed to the design of the study, conducted part of Experiment 1, conducted Experiment 2, was responsible for overall data analysis, drafted the manuscript. V.C.S.d.l.B. contributed to the design of the study, conducted Experiment 1, contributed to data analysis. D.V. designed and supervised the study and critically revised the manuscript.

Competing interests. We declare we have no competing interests.

Funding. This work was supported by the Engineering and Physical Sciences Research Council (grant no. EP/M506631/1). It was part of the grant submitted by D.V., which covered PhD scholarship for G.Z.

Acknowledgements. No contributions from other people were made.

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
