## [Reviewer comments · Royal Society Open Science]

Review History

RSOS-200955.R0 (Original submission)

Review form: Reviewer 1

Is the manuscript scientifically sound in its present form?

Yes

Are the interpretations and conclusions justified by the results?

Yes

Is the language acceptable?

Yes

Do you have any ethical concerns with this paper?

No

Have you any concerns about statistical analyses in this paper?

No

Recommendation?

Accept with minor revision (please list in comments)

Comments to the Author(s)

This study shows clearly that, while there are some small differences, the effects of pictorial depth cues on depth judgements in pictures are very similar for people with strabismus and the general population. The study is very well performed and presented. In particular, great care has been taken in framing the participants' task (whether judgements in the picture plane or in pictorial space), explaining this to participants and making direct comparisons. This is not always the case in the literature, leaving readers (and more importantly participants) unclear how the task should be performed. The broad lack of a difference between the two groups is, in this case, compelling – since both groups show the same effects of stimulus information and task. The motivation of the study is clear, the methods detailed, and refined across the two studies, and the results are clear and informative. I have only the following minor comments to add:

Page 5 line 7 delete 'to'

Pag 5 lines 24 – it is stated that binocular cues (disparity) provide only relative information; this is a bit unclearly worded, excluding the binocular cues of convergence; also absolute depth can be extracted if (image based) calibration is achieved.

Please provide some more information about the staircase used – in particular what were the rules for selecting new stimuli?

As a note, I had a number of methodological questions about the first experiment, related to the use of both staircase and adjustment methods. I removed these as they were covered in detail by the authors, so thank you for the clarity and critical self-assessment.

How are the results determined for the staircase conditions (is the result for each condition the mean of reversal points, for example?)

Review form: Reviewer 2

Is the manuscript scientifically sound in its present form?

Yes

Are the interpretations and conclusions justified by the results?

Yes

Is the language acceptable?

Yes

Do you have any ethical concerns with this paper?

No

Have you any concerns about statistical analyses in this paper?

No

Recommendation?

Major revision is needed (please make suggestions in comments)

Comments to the Author(s)

See attached (Appendix A).

Review form: Reviewer 3

Is the manuscript scientifically sound in its present form?

No

Are the interpretations and conclusions justified by the results?

Yes

Is the language acceptable?

Yes

Do you have any ethical concerns with this paper?

No

Have you any concerns about statistical analyses in this paper?

Yes

Recommendation?

Major revision is needed (please make suggestions in comments)

Comments to the Author(s)

For Authors:

General comment:

While the overall study question is interesting, the manuscript needs improvement in different areas. The study proposed to explore impairment of relative depth in Strabismic individuals however the inclusion and screening criteria are not clear. While strabismus generally results in loss of stereo-acuity, some types of strabismus such as intermittent exotropia retain residual stereo-acuity.¹ Similarly the study includes participant with phoria in the group with "limited stereoacuity " but in individual with phorias stereo-acuity is generally good. Also amblyopes are included in the study, are these with strabismus ? one is stated as phoria. The study needs to be clear on the inclusion criteria and report on the individual levels of stereoacuity and what is considered as criteria for "limited stereoacuity". In page 7 it is reported that participants were selected on the "self-reported occasional double vision and self-reported inability to use 3D technology", it would be good to include participants based on results of clinical tests rather than self-reported symptoms.

The study and its results would be more robust with addition of more participants with different conditions such as Strabismus, Intermittent Strabismus, Amblyopia and analysis of these subgroups. Also addition of clinical data such as type of strabismus, amblyopia, level of stereoacuity for the participants will be useful.

Specific Comments:

Summary/abstract: Needs descriptive data and results of statistical tests while describing the results.

Introduction:

Page 1, Line 55: "Due to a range of neurological or physiological issues" - provide few examples of these.

Page 2, Line 12 - 13: needs amending; not all double vision results in amblyopia, there are other sensory adaptation such as Eccentric fixation, Abnormal retinal correspondence.

Page 2, Line 24: Popular media accounts combined . . . , provide reference for these.

Figure 1: describe what the values in the figure are.

Page 7, line 3 – 4: "Amblyopia onset is most often associated with strabismus in early .." – the last portion of sentence is inaccurate, the onset of amblyopia is during the childhood only, the development of strabismus in adulthood results in diplopia and not amblyopia.

Page 13, Table 1: Provide further details such as St. dev for age, if stereo-acuity was measured report these along with the results of cover test. What were the types of amblyopia?

Page 13, line 25: Provide details of the ANOVA. Did the data satisfy the assumptions for these tests, such as normality, variance etc. considering the small sample size and unequal number of participants (8 and 20 in target and control group).

Page 15, Line 22 – 27: "It is plausible that this similarity is due to a methodological issue, rather than actual lack of perceptual differentiation between the 2D and 3D judgments. Specifically, it is possible that the order of testing, the 2D task followed by the 3D task, may have caused participants to misinterpret the initial 2D task to be a 3D task"

- The order of task could have been randomised. Please consider this if recruiting more participants.

Page 16, Line 35: 5.2 Task and procedure: What is the rational behind using 2AFC and method of adjustment? would be useful to describe this here.

Page 18, Line 11 – 12: 5.4 Participants: "14 individuals with no or "

- Were these different from the participants from the first experiment which only had 8.

- Provide participant details from clinical tests such as stereoacuity, cover test in a Table here,

Page 20, Line 43- : "For the 2AFC task, a 2x3x2 mixed design ANOVAs" Report if the data satisfies the assumption for the ANOVA such as Normality, equality of variance etc.

Discussion

Page 25, line 20: Use statistical term rather than slightly, was the difference significant, if not report it as such or justify as a trend.

1 Stathacopoulos RA, Rosenbaum AL, Zanoni D et al. Distance Stereoacuity: Assessing Control in Intermittent Exotropia. *Ophthalmology* 1993; 100: 495-500.

Decision letter (RSOS-200955.R0)

Dear Dr Zlatkute,

The Editors assigned to your paper RSOS-200955 "Unimpaired perception of relative depth from perspective cues in strabismus" have now received comments from reviewers and would like you to revise the paper in accordance with the reviewer comments and any comments from the Editors. Please note this decision does not guarantee eventual acceptance.

Please submit your revised manuscript and required files (see below) no later than 21 days from today's (ie 13-Aug-2020) date. Note: the ScholarOne system will 'lock' if submission of the revision is attempted 21 or more days after the deadline. If you do not think you will be able to meet this deadline please contact the editorial office immediately.

Best regards,

on behalf of Dr Isabelle Mareschal (Associate Editor) and Essi Viding (Subject Editor)
openscience@royalsociety.org

Associate Editor Comments to Author (Dr Isabelle Mareschal):

Expert reviewers have provided feedback on your submission. Please provide a point by point reply to their queries, notably providing a clearer theoretical motivation for the study and description of the inclusion criteria for the strabismic group.

Reviewer comments to Author:

Reviewer: 1
Comments to the Author(s)

This study shows clearly that, while there are some small differences, the effects of pictorial depth cues on depth judgements in pictures are very similar for people with strabismus and the general population. The study is very well performed and presented. In particular, great care has been taken in framing the participants' task (whether judgements in the picture plane or in pictorial space), explaining this to participants and making direct comparisons. This is not always the case in the literature, leaving readers (and more importantly participants) unclear how the task should be performed. The broad lack of a difference between the two groups is, in this case, compelling –

since both groups show the same effects of stimulus information and task. The motivation of the study is clear, the methods detailed, and refined across the two studies, and the results are clear and informative. I have only the following minor comments to add:

Page 5 line 7 delete 'to'

Pag 5 lines 24 – it is stated that binocular cues (disparity) provide only relative information; this is a bit unclearly worded, excluding the binocular cues of convergence; also absolute depth can be extracted if (image based) calibration is achieved.

Please provide some more information about the staircase used – in particular what were the rules for selecting new stimuli?

As a note, I had a number of methodological questions about the first experiment, related to the use of both staircase and adjustment methods. I removed these as they were covered in detail by the authors, so thank you for the clarity and critical self-assessment.

How are the resulted determined for the staircase conditions (is the result for each condition the mean of reversal points, for example?)

Reviewer: 2

Comments to the Author(s)

See attached.

Reviewer: 3

Comments to the Author(s)

For Authors:

General comment:

While the overall study question is interesting, the manuscript needs improvement in different areas. The study proposed to explore impairment of relative depth in Strabismic individuals however the inclusion and screening criteria are not clear. While strabismus generally results in loss of stereo-acuity, some types of strabismus such as intermittent exotropia retain residual stereo-acuity.¹ Similarly the study includes participant with phoria in the group with "limited stereoacuity " but in individual with phorias stereo-acuity is generally good. Also amblyopes are included in the study, are these with strabismus ? one is stated as phoria. The study needs to be clear on the inclusion criteria and report on the individual levels of stereoacuity and what is considered as criteria for "limited stereoacuity". In page 7 it is reported that participants were selected on the "self-reported occasional double vision and self-reported inability to use 3D technology", it would be good to include participants based on results of clinical tests rather than self-reported symptoms.

The study and its results would be more robust with addition of more participants with different conditions such as Strabismus, Intermittent Strabismus, Amblyopia and analysis of these subgroups. Also addition of clinical data such as type of strabismus, amblyopia, level of stereoacuity for the participants will be useful.

Specific Comments:

Summary/abstract: Needs descriptive data and results of statistical tests while describing the results.

Introduction:

Page 1, Line 55: "Due to a range of neurological or physiological issues" - provide few examples of these.

Page 2, Line 12 - 13: needs amending; not all double vision results in amblyopia, there are other sensory adaptation such as Eccentric fixation, Abnormal retinal correspondence.

Page 2, Line 24: Popular media accounts combined ..., provide reference for these.

Figure 1: describe what the values in the figure are.

Page 7, line 3 - 4: "Amblyopia onset is most often associated with strabismus in early .." - the last portion of sentence is inaccurate, the onset of amblyopia is during the childhood only, the development of strabismus in adulthood results in diplopia and not amblyopia.

Page 13, Table 1: Provide further details such as St. dev for age, if stereo-acuity was measured report these along with the results of cover test. What were the types of amblyopia?

Page 13, line 25: Provide details of the ANOVA. Did the data satisfy the assumptions for these tests, such as normality, variance etc. considering the small sample size and unequal number of participants (8 and 20 in target and control group).

Page 15, Line 22 - 27: "It is plausible that this similarity is due to a methodological issue, rather than actual lack of perceptual differentiation between the 2D and 3D judgments. Specifically, it is possible that the order of testing, the 2D task followed by the 3D task, may have caused participants to misinterpret the initial 2D task to be a 3D task"
- The order of task could have been randomised. Please consider this if recruiting more participants.

Page 16, Line 35: 5.2 Task and procedure: What is the rational behind using 2AFC and method of adjustment? would be useful to describe this here.

Page 18, Line 11 - 12: 5.4 Participants: "14 individuals with no or "
- Were these different from the participants from the first experiment which only had 8.
- Provide participant details from clinical tests such as stereoacuity, cover test in a Table here,
Page 20, Line 43- : "For the 2AFC task, a 2x3x2 mixed design ANOVAs" Report if the data satisfies the assumption for the ANOVA such as Normality, equality of variance etc.

Discussion

Page 25, line 20: Use statistical term rather than slightly, was the difference significant, if not report it as such or justify as a trend

1 Stathacopoulos RA, Rosenbaum AL, Zanoni D et al. Distance Stereoacuity: Assessing Control in Intermittent Exotropia. *Ophthalmology* 1993; 100: 495-500.

===PREPARING YOUR MANUSCRIPT===

Your revised paper should include the changes requested by the referees and Editors of your manuscript. You should provide two versions of this manuscript and both versions must be provided in an editable format:
one version identifying all the changes that have been made (for instance, in coloured highlight, in bold text, or tracked changes);

a 'clean' version of the new manuscript that incorporates the changes made, but does not highlight them. This version will be used for typesetting if your manuscript is accepted. Please ensure that any equations included in the paper are editable text and not embedded images.

===PREPARING YOUR REVISION IN SCHOLARONE===

Author's Response to Decision Letter for (RSOS-200955.R0)

See Appendix B.

RSOS-200955.R1 (Revision)

Review form: Reviewer 2

Is the manuscript scientifically sound in its present form?

Yes

Are the interpretations and conclusions justified by the results?

Yes

Is the language acceptable?

Yes

Do you have any ethical concerns with this paper?

No

Have you any concerns about statistical analyses in this paper?

No

Recommendation?

Accept with minor revision (please list in comments)

Comments to the Author(s)

The manuscript is clearer and improved as a result of the changes. Two minor comments:

My understanding of the new text added on p5, l42 is that the authors predict poorer sensitivity (greater variability) in judgements of monocular slant because of the lack of a bottom-up binocular disparity signal with which to correlate the pictorial cues. If so, it would help to make the prediction explicit.

The previous citation in my review was correct and unambiguous (Koenderink, J. J., van Doorn, A. J., & Kappers, A. M. (1995). Depth relief. *Perception*, 24, 115-126). The authors say in the Introduction: "Notice that this effect is the exact opposite of that for pictorial relief: closing an eye deepens pictorial relief, but flattens 'real' relief.", stating this as if it were accepted wisdom. Indeed, it makes perfect sense. The data in Fig 7 back up the statement convincingly. Viewing a photograph binocularly is the equivalent of their synoptic condition in that there is a zero disparity signal across the image rather than an absence of a disparity signal (monocular). The authors should discuss this evidence and interpretation (which is, after all compatible with cue combination theory) rather than ignoring it: "previous work has shown no influence of conflicting binocular disparity information in 3D shape judgements...(37-40)".

Review form: Reviewer 3

Is the manuscript scientifically sound in its present form?

Yes

Are the interpretations and conclusions justified by the results?

Yes

Is the language acceptable?

Yes

Do you have any ethical concerns with this paper?

No

Have you any concerns about statistical analyses in this paper?

No

Recommendation?

Accept with minor revision (please list in comments)

Comments to the Author(s)

Refer to the attached document (Appendix C).

Decision letter (RSOS-200955.R1)

Dear Dr Zlatkute

On behalf of the Editors, we are pleased to inform you that your Manuscript RSOS-200955.R1 "Unimpaired perception of relative depth from perspective cues in strabismus" has been accepted for publication in Royal Society Open Science subject to minor revision in accordance with the referees' reports. Please find the referees' comments along with any feedback from the Editors below my signature.

Please submit your revised manuscript and required files (see below) no later than 7 days from today's (ie 05-Nov-2020) date. Note: the ScholarOne system will 'lock' if submission of the revision is attempted 7 or more days after the deadline. If you do not think you will be able to meet this deadline please contact the editorial office immediately.

Kind regards,

Anita Kristiansen
Editorial Coordinator

on behalf of Dr Isabelle Mareschal (Associate Editor) and Essi Viding (Subject Editor)
openscience@royalsociety.org

Associate Editor Comments to Author (Dr Isabelle Mareschal):

Comments to the Author:

Both reviewers' previous concerns have been addressed. Please address the few remaining queries, directly in the manuscript and in your replies.

Reviewer comments to Author:
Reviewer: 3

Comments to the Author(s)
Refer to the attached document.

Reviewer: 2

Comments to the Author(s)
The manuscript is clearer and improved as a result of the changes. Two minor comments:

My understanding of the new text added on p5, l42 is that the authors predict poorer sensitivity (greater variability) in judgements of monocular slant because of the lack of a bottom-up

binocular disparity signal with which to correlate the pictorial cues. If so, it would help to make the prediction explicit.

The previous citation in my review was correct and unambiguous (Koenderink, J. J., van Doorn, A. J., & Kappers, A. M. (1995). Depth relief. *Perception*, 24, 115-126). The authors say in the Introduction: "Notice that this effect is the exact opposite of that for pictorial relief: closing an eye deepens pictorial relief, but flattens 'real' relief.", stating this as if it were accepted wisdom. Indeed, it makes perfect sense. The data in Fig 7 back up the statement convincingly. Viewing a photograph binocularly is the equivalent of their synoptic condition in that there is a zero disparity signal across the image rather than an absence of a disparity signal (monocular). The authors should discuss this evidence and interpretation (which is, after all compatible with cue combination theory) rather than ignoring it: "previous work has shown no influence of conflicting binocular disparity information in 3D shape judgements...(37-40)".

===PREPARING YOUR MANUSCRIPT===

===PREPARING YOUR REVISION IN SCHOLARONE===

Author's Response to Decision Letter for (RSOS-200955.R1)

See Appendix D.

Decision letter (RSOS-200955.R2)

Dear Dr Zlatkute,

It is a pleasure to accept your manuscript entitled "Unimpaired perception of relative depth from perspective cues in strabismus" in its current form for publication in Royal Society Open Science.

We note that the contact information for one of your co-authors appears to be incorrect. Please can you check the following email address, and reply to this email with the correct and updated contact:

- vanessa.sagnay_de_la_bastisa@kcl.ac.uk

Best regards,

on behalf of Dr Isabelle Mareschal (Associate Editor) and Essi Viding (Subject Editor)
openscience@royalsociety.org

Appendix A

This paper examines the extent to which width judgements are made in 2D (in the picture plane) or a perceived 3D plane for participants who have limited binocular vision. They compare the results with those found for control participants and find (essentially) no difference. The studies are well done and for the most part well described. The main issue is whether there is a compelling case to document this negative result.

1. In the current manuscript, there is very little to persuade the reader that strabismic or other participants with poor stereoacuity are likely to see pictorial monocular depth cues differently from control participants. The authors quote anecdotal reports (p5,147) and claim, without making any clear argument, that: 'The capacity for strabismic observers to perceive depth ... may remain impoverished'. In the Discussion (p25,158) the authors talk about 'the use of the primary static monocular cue to depth perception (perspective convergence) [being] compromised in strabismus'. If 'impoverished' or 'compromised' depth perception means, in this context, a regression towards the 2D interpretation, then an expectation of the reverse result is supported by the literature the authors cite (eg p25,130) and others (eg Koenderink, van Doorn, Kappers, 1995), i.e that binocular viewing of a picture should give rise to a flatter interpretation. Indeed, the authors find some evidence for this (p25,120). So, it is not clear whether the authors expect the strabismic participants to show a difference from controls towards the direction of a 2D or a 3D interpretation, nor is the theoretical case for a given position made clear. In photographs, pictorial cues for a certain slant are powerful and one might expect the constraints that these cues provide to dominate independent of whether the scene is viewed monocularly or binocularly (or by strabismic or control participants). In any case, the justification in the manuscript for making this comparison between strabismic and control participants is slim.

Minor

2. The authors do not justify why they tested the participants only with binocular viewing. Do they predict a difference between the two eyes for strabismic participants? Do they predict a change in bias for control participants when they switch from binocular to monocular viewing? This would form an interesting comparison with the strabismic participants because it would enable one to tease apart the binocular/monocular effect from the strabismic/control participant effect. At least, it would if any effect had been found. At a minimum, the authors should justify their decision to use only binocular presentation.

3. p12, 141. 'Control' has two meanings: control participant and control condition. There are lots of abbreviations used, too (2D CP etc). It would be good if the authors could use different terms to avoid a headache for the reader.

4. p18, 137 An author was excluded from Expt 2 to avoid potential bias but not from Expt 1. This sounds odd.

5. p26, 112-27 This literature shows that when binocular cues are important strabismics are impaired but when these cues are unimportant, strabismics are unimpaired. This supports point 1 above.

6. Table 1. Corrected strabismus and amblyopia are not mutually exclusive. Clarify.

Appendix B

Reviewer: 1

This study shows clearly that, while there are some small differences, the effects of pictorial depth cues on depth judgements in pictures are very similar for people with strabismus and the general population. The study is very well performed and presented. In particular, great care has been taken in framing the participants' task (whether judgements in the picture plane or in pictorial space), explaining this to participants and making direct comparisons. This is not always the case in the literature, leaving readers (and more importantly participants) unclear how the task should be performed. The broad lack of a difference between the two groups is, in this case, compelling – since both groups show the same effects of stimulus information and task. The motivation of the study is clear, the methods detailed, and refined across the two studies, and the results are clear and informative. I have only the following minor comments to add:

We thank the reviewer for their positive appraisal.

Page 5 line 7 delete 'to'

Done

Page 5 lines 24 – it is stated that binocular cues (disparity) provide only relative information; this is a bit unclearly worded, excluding the binocular cues of convergence; also absolute depth can be extracted if (image based) calibration is achieved.

We have changed this to the following:

“binocular disparities specify relative depth relations and need to be scaled by egocentric distance cues such as convergence in order to derive scaled (absolute) depth.”

Please provide some more information about the staircase used – in particular what were the rules for selecting new stimuli?

This information has now been added in the methods section for each experiment (Experiment 1 page 15 “stimuli tested” section; Experiment 2 page 23 (line 4)).

As a note, I had a number of methodological questions about the first experiment, related to the use of both staircase and adjustment methods. I removed these as they were covered in detail by the authors, so thank you for the clarity and critical self-assessment.

We thank the reviewer for spotting this omission in details on the staircase procedure and have added more detail on the psychophysical methods in both experiments (see page 15 and 23)

How are the results determined for the staircase conditions (is the result for each condition the mean of reversal points, for example?)

Yes, they were averages of values at reversal points. We have now added this information in stimuli tested section for each experiment (Experiment 1 page 15 “stimuli tested” section; Experiment 2 page 23 (lines 7-8)).

Reviewer: 2

This paper examines the extent to which width judgements are made in 2D (in the picture plane) or a perceived 3D plane for participants who have limited binocular vision. They compare the results with those found for control participants and find (essentially) no difference. The studies are well done and for the most part well described. The main issue is whether there is a compelling case to document this negative result.¹ In the current manuscript, there is very little to persuade the reader that strabismic or other participants with poor stereoacuity are likely to see pictorial monocular depth cues differently from control participants. The authors quote anecdotal reports (p5,147) and claim, without making any clear argument, that: 'The capacity for strabismic observers to perceive depth ... may remain impoverished'. In the Discussion (p25,158) the authors talk about 'the use of the primary static monocular cue to depth perception (perspective convergence) [being] compromised in strabismus'. If 'impoverished' or 'compromised' depth perception means, in this context, a regression towards the 2D interpretation, then an expectation of the reverse result is supported by the literature the authors cite (eg p25,130) and others (eg Koenderink, van Doorn, Kappers, 1995), i.e that binocular viewing of a picture should give rise to a flatter interpretation. Indeed, the authors find some evidence for this (p25,120). So, it is not clear whether the authors expect the strabismic participants to show a difference from controls towards the direction of a 2D or a 3D interpretation, nor is the theoretical case for a given position made clear. In photographs, pictorial cues for a certain slant are powerful and one might expect the constraints that these cues provide to dominate independent of whether the scene is viewed monocularly or binocularly (or by strabismic or control participants). In any case, the justification in the manuscript for making this comparison between strabismic and control participants is slim.

We thank the reviewer for pointing out the lack of clarity regarding predictions for perceived pictorial depth under monocular and binocular viewing. We have now added text in the introduction on Page 6 and 7, which we hope addresses more clearly the problem. Note that the paper the reviewer refers to (Koenderink, van Doorn, 1995) is for judgements on a real sphere, not pictured objects. With respect to Koenderink et al.1994 which was on pictures, we note that they are the only researchers to show a difference in relative depth judgements between monocular and binocular viewing of pictured objects. Moreover, the results are complicated by the fact it involved a highly localised task (gauge figure) and they only tested 2-3 observers who were all authors and therefore familiar with the study. Other studies that have tested naïve subjects on tasks ranging from global shape judgement, surface slant and relative size tasks have not found any difference (Erkelens, 2015; Vishwanath & Hibbard, 2013; Wijntjes et al., 2012; Cooper & Banks, 2012). While in our study we did find a small difference between monocular and binocular viewing, the task is one that has both a 2D and 3D task interpretation and therefore the small differences (especially in light of previous studies) needs to be interpreted with caution (we have addressed this in discussion page 35)

With respect to the more general comment regarding the motivation for the study, we note the primary aim of the studies was to understand the perception of relative depth from monocular cues in strabismics (not differences between monocular and binocular viewing). Prior literature has not addressed the question of relative depth from monocular cues in strabismics and there is significant confusion among the general public regarding what sorts of depth strabismics can and cannot perceive (e.g., we have received scores of queries from

the general public over the years pointing to a confusion among strabismics about how their vision compares to those with normal binocular vision). This is highlighted in the most widely read popular scientific account of strabismus and depth perception to date (Barry, 2009) where the claim is that strabismics do not perceive relative depth from monocular cues in the same way as non-strabismics do—implying either that monocular cues do not provide a bottom-up perception of depth, or that they somehow rely on the presence of binocular disparity. This view is also endorsed by other researchers (e.g., Rogers, 2019) where the claim is that pictorial cues do not provide quantitative depth perception in the way that binocular disparity does. While some vision researchers would predict that strabismus should make no difference to depth perception from monocular cues (or pictorial depth), there are many other vision researchers who were surprised that we did not find any difference between the two groups. So, we don't agree that the question is a settled or trivial one. We believe there is a compelling case to empirically answer this question, both from a basic science point of view and in the wider public interest.

Minor

2. The authors do not justify why they tested the participants only with binocular viewing. Do they predict a difference between the two eyes for strabismic participants? Do they predict a change in bias for control participants when they switch from binocular to monocular viewing? This would form an interesting comparison with the strabismic participants because it would enable one to tease apart the binocular/monocular effect from the strabismic/control participant effect. At least, it would if any effect had been found. At a minimum, the authors should justify their decision to use only binocular presentation.

We are not sure what the reviewer is referring to here. Experiment 2 does exactly what the reviewer has suggested—i.e., it tests both monocular and binocular presentation.

3. p12, l41. 'Control' has two meanings: control participant and control condition. There are lots of abbreviations used, too (2D CP etc). It would be good if the authors could use different terms to avoid a headache for the reader.

We thank the reviewer for pointing out this source of confusion. We have changed “control participants” to “normal stereovision participants” throughout the manuscript to avoid this confusion.

4. p18, l37 An author was excluded from Expt 2 to avoid potential bias but not from Expt 1. This sounds odd.

The first author's result was included in the first study because the experiment was set up and conducted by the second author, while in Experiment 2, the first author tested herself. However, to remove any doubt, we have eliminated the author from the Experiment 1 as well. We carried out statistical tests again without the author which showed no notable change; we have updated the values and Figure 7.

5. p26, l12-27 This literature shows that when binocular cues are important strabismics are impaired but when these cues are unimportant, strabismics are unimpaired. This supports point 1 above.

We believe that this argument (point 1 above) could also have been made to claim that there was no compelling case for the researchers who conducted the studies referenced in this paragraph to have done so, since a thought experiment would have sufficed. But this would mean there would be virtually no literature on depth perception in strabismics, and an understanding of depth perception in strabismics would be left to anecdotal reports such as (Barry, 2009), which is a moving story with significant scientific implications, but where some of the claims are not supported by the empirical data, as we have shown here.

While the present results can be argued as being predicted under certain assumptions of the nature and developmental implementation of monocular cues, they are surprising under other assumptions prevalent in the field. Our study, we believe, is therefore a significant addition to the literature in giving a fuller picture of depth perception in strabismus and bears upon the broader debate on the developmental trajectory of depth cues and the nature of differences between monocular and binocular depth perception.

6. Table 1. Corrected strabismus and amblyopia are not mutually exclusive. Clarify.

We had originally put all vision test details of the limited/no stereovision cohort in tables in supplementary material. We have now moved this to the main manuscript (Tables 1 and 2, pages 17 and 25-26) and clarified this further in text (page 9, 10). We hope this addresses the reviewer's concern.

Reviewer: 3

Comments to the Author(s)

For Authors:

General comment:

While the overall study question is interesting, the manuscript needs improvement in different areas. The study proposed to explore impairment of relative depth in Strabismic individuals however the inclusion and screening criteria are not clear. While strabismus generally results in loss of stereo-acuity, some types of strabismus such as intermittent exotropia retain residual stereo-acuity.1 Similarly the study includes participant with phoria in the group with "limited stereoacuity " but in individual with phorias stereo-acuity is generally good. Also amblyopes are included in the study, are these with strabismus ? one is stated as phoria. The study needs to be clear on the inclusion criteria and report on the individual levels of stereoacuity and what is considered as criteria for "limited stereoacuity". In page 7 it is reported that participants were selected on the "self-reported occasional double vision and self-reported inability to use 3D technology", it would be good to include participants based on results of clinical tests rather than self-reported symptoms.

We have moved the detailed table showing the results from all the tests for the limited/no stereovision group into the main manuscript (Tables 1 and 2, pages 17 and 25-26) and have provided more details in text on the recruitment and inclusion criteria for limited stereovision (pages 9-11).

The study and its results would be more robust with addition of more participants with different

conditions such as Strabismus, Intermittent Strabismus, Amblyopia and analysis of these subgroups. Also addition of clinical data such as type of strabismus, amblyopia, level of stereoacuity for the participants will be useful.

We agree with the reviewer that further studies to examine these and associated question on a bigger cohort and in a clinical setting would be useful. However, we also note that recruitment for these conditions for experimental testing remains a challenge, not only in a non-clinical context but also in a clinical setting. In a previous study in which we collaborated with a school of optometry in the US, only 7-8 clinical volunteers were able to be recruited there over the span of 6 months. While a larger cohort would have been preferable, we think that the pattern of the data do not suggest that any theoretically interesting differences related to the broader findings are likely to emerge from testing a larger cohort; notwithstanding the fact that a larger cohort could potentially yield information on more fine-grained differences based on specific impairments.

Specific Comments:

Summary/abstract: Needs descriptive data and results of statistical tests while describing the results.

We have added the following to the abstract:

The biasing effect of perspective information on the 2D judgements (3D cue susceptibility) was highly significant and comparable for both subject groups in both the psychophysical tasks (all p 's < 0.001) with no statistically significant difference found between the two groups. Both groups showed an underestimation in the 3D task with no significant difference between the group's judgements in the 2AFC task, but a small statistically significant difference (ratio difference of approx. 10%, $p=0.016$) in Method of Adjustment task. A small but significant effect of viewing condition (monocular vs. binocular) was revealed only in the non-strabismic group (ratio difference of approx. 6%, $p=0.002$)

Introduction:

Page 1, Line 55: "Due to a range of neurological or physiological issues" – provide few examples of these.

Examples given on page 3 (line 9-10).

Page 2, Line 12 - 13: needs amending; not all double vision results in amblyopia, there are other sensory adaptation such as Eccentric fixation, Abnormal retinal correspondence.

We have added amended the statement on page 3 (lines 25-31).

Page 2, Line 24: Popular media accounts combined ..., provide reference for these.

We have added the references to these on page 3 (line 44).

Figure 1: describe what the values in the figure are.

We have now indicated this in the figure.

Page 7, line 3 – 4: "Amblyopia onset is most often associated with strabismus in early .." – the last portion of sentence is inaccurate, the onset of amblyopia is during the childhood only, the development of strabismus in adulthood results in diplopia and not amblyopia.

We have clarified this on page 9 (line 20).

Page 13, Table 1: Provide further details such as St. dev for age, if stereo-acuity was measured report these along with the results of cover test. What were the types of amblyopia?

We had put all these details in supplementary materials but have now moved the table to the main manuscript (Table 1) and also added age averages to main text page 16 (lines 16 and 27) for Experiment 1 and page 24 (line 16) for Experiment 2.

Page 13, line 25: Provide details of the ANOVA. Did the data satisfy the assumptions for these tests, such as normality, variance etc. considering the small sample size and unequal number of participants (8 and 20 in target and control group).

We have now added more details page 18.

Page 15, Line 22 – 27: "It is plausible that this similarity is due to a methodological issue, rather than actual lack of perceptual differentiation between the 2D and 3D judgments. Specifically, it is possible that the order of testing, the 2D task followed by the 3D task, may have caused participants to misinterpret the initial 2D task to be a 3D task"

The order of task could have been randomised. Please consider this if recruiting more participants.

The issue of task order was discussed in the discussion section of Experiment 1 on page 20 (line 33-38) where the quote above was taken from) and the order was changed in experiment 2. We do not believe that a randomized order is the correct approach given the ambiguity in the task and the necessity that subjects get a clear understanding of what they are supposed to do. It is clear that the change in task order in experiment 2 did make a difference in the predicted direction. Subjects in experiment 2, after we changed order, were more clearly differentiating between the 2D and 3D task (see Figure 12 in Supplementary materials).

More generally, due to the nature of the stimuli and tasks, the stimulus and task order were selected to reduce any confound of stimulus presentation and/or confusion of the 3D vs 2D task. For example it is important that the 2D- control perspective condition (converging lines only stimulus) be tested before the Pictorial Perspective condition, in order to get the baseline effect. Doing it in the reverse order opens it up to a possible confound that the subject interprets the lines as receding in space simply because they saw the pictorial version of it previously. Page 22 (lines 1-21) provide the detailed justification for the order that we chose.

Page 16, Line 35: 5.2 Task and procedure: What is the rationale behind using 2AFC and method of adjustment? would be useful to describe this here.

We have provided the rationale on page 21 (lines 41-50).

Page 18, Line 11 – 12: 5.4 Participants: "14 individuals with no or "

- Were these different from the participants from the first experiment which only had 8.

- Provide participant details from clinical tests such as stereoacuity, cover test in a Table here,

The subjects tested in both experiment 1 and 2 have been clarified on page 23 (lines 33-37) and in the detailed visual information tables (Tables 1 and 2, pages 17 and 25-26), which also contain all the information on vision and stereoacuity tests

Page 20, Line 43- : "For the 2AFC task, a 2x3x2 mixed design ANOVAs" Report if the data satisfies the assumption for the ANOVA such as Normality, equality of variance etc.

We have now added more details on pages 30-31.

Discussion

Page 25, line 20: Use statistical term rather than slightly, was the difference significant, if not report it as such or justify as a trend.

We have removed the word “slightly” and rephrased this. See page 35 lines 18-31.

1 Stathacopoulos RA, Rosenbaum AL, Zanoni D et al. Distance Stereoacuity: Assessing Control in Intermittent Exotropia. *Ophthalmology* 1993; 100: 495-500.

- this reference is linked to the first comment pointing out that just because a person has strabismus he/she does not automatically have impaired stereoacuity. This is also addressed in the more detailed participant visual information tables (Table 1 and 2, pages 17 and 25-26) and the additional information about recruitment criteria pages 9-11.

Appendix C

The manuscript has improved with the consideration of comments from the reviewer. I have some minor comments summarised below:

Introduction:

Page 1, Line 55: "Due to a range of neurological or physiological issues" – provide few examples of these.

Examples given on page 3 (line 9-10).

Authors have added the following:

"physiological issues ranging from astigmatism to developmental delays and also genetic conditions as Down's syndrome"

- Remove astigmatism as this alone is not causative for Strabismics being unable to fixate the two eyes on a single target point in space.

Page 7, line 3 – 4: "Amblyopia onset is most often associated with strabismus in early .." – the last portion of sentence is inaccurate, the onset of amblyopia is during the childhood only, the development of strabismus in adulthood results in diplopia and not amblyopia. We have clarified this on page 9 (line 20).

Authors have added the following:

"Amblyopia onset is most often associated with strabismus in early childhood and intermitted strabismus is often observed among amblyopes in adulthood."

- Do author mean to say the strabismus persists into adulthood in amblyopia? The last part of the sentence is unclear, as amblyopes could have both constant as well as intermittent strabismus in adulthood or in case of refractive amblyopia, no manifest deviation may also occur.

Page 13, line 25: Provide details of the ANOVA. Did the data satisfy the assumptions for these tests, such as normality, variance etc. considering the small sample size and unequal number of participants (8 and 20 in target and control group).

We have now added more details page 18.

- Report Mauchly's Tests of Sphericity prior to the main results.

Other Minor comments:

- Add units to age (years)
- Use appropriate units for stereoacuity. In the manuscript it is recorded as e.g. 800' to 40' arcsecs. The Unit for second of arc is: ". Using just one either " or arcsecs, would be adequate once introduced at start.

Appendix D

We thank the editor for accepting our manuscript for publication in RSOS. Below we have responded (black) to the reviewer minor comments (in blue).

Reviewer: 2

Comments to the Author(s)

The manuscript is clearer and improved as a result of the changes. Two minor comments:

My understanding of the new text added on p5, l42 is that the authors predict poorer sensitivity (greater variability) in judgements of monocular slant because of the lack of a bottom-up binocular disparity signal with which to correlate the pictorial cues. If so, it would help to make the prediction explicit.

We do not predict anything specific regarding variability of monocular depth cues in that statement. We simply highlight the developmental argument that the efficacy of monocular depth cues may depend on correlating/calibrating monocular cue patterns with depth derived directly from disparity mechanisms, and therefore, that monocular depth cues may not be as effective in strabismics as it is in typical develop binocular observers in inferring depth.

The previous citation in my review was correct and unambiguous (Koenderink, J. J., van Doorn, A. J., & Kappers, A. M. (1995). Depth relief. *Perception*, 24, 115-126). The authors say in the Introduction: "Notice that this effect is the exact opposite of that for pictorial relief: closing an eye deepens pictorial relief, but flattens 'real' relief.", stating this as if it were accepted wisdom. Indeed, it makes perfect sense. The data in Fig 7 back up the statement convincingly. Viewing a photograph binocularly is the equivalent of their synoptic condition in that there is a zero disparity signal across the image rather than an absence of a disparity signal (monocular). The authors should discuss this evidence and interpretation (which is, after all compatible with cue combination theory) rather than ignoring it: "previous work has shown no influence of conflicting binocular disparity information in 3D shape judgements...(37-40)".

We are unclear on the reviewer's argument that the synoptic condition is the same as viewing a picture binocularly ("Viewing a photograph binocularly is the equivalent of their synoptic condition") The synopter abolishes disparity throughout the visual field (zero disparity) while binocular viewing of a picture has non-zero horizontal and vertical disparities that specify the frontoparallel orientation of the picture surface. Moreover, Koenderink et als. highlight precisely that these two conditions are optically and perceptually different—synoptic viewing yields a more vivid impression of depth than binocular viewing.

Because the Koenderink paper compares synoptic and binocular viewing, rather than monocular and binocular viewing, it is not directly relevant to the point made in our paper. It is true that the results can be interpreted as compatible with a cue-combination account of depth in pictures, but given the special aspects of their results, we feel discussing them is outside the scope of our paper. [the koenderink study involves synoptic vs binocular viewing, not monocular vs binocular viewing; only tested non-naïve authors using a specific local task (guage figure) and those results have not been replicated by any other researchers; the results of all studies since then comparing monocular and binocular viewing of pictures using different tasks (at least 4) contradict the cue integration account—even results from researchers who promote the cue-integration theory: Marty Banks group]. These aspects have been addressed in detail elsewhere (Vishwanath, 2014).

Reviewer: 3

The manuscript has improved with the consideration of comments from the reviewer. I have some minor comments summarised below:

Introduction: Page 1, Line 55: "Due to a range of neurological or physiological issues" – provide few examples of these.

Authors: Examples given on page 3 (line 9-10).

Authors have added the following: "physiological issues ranging from astigmatism to developmental delays and also genetic conditions as Down's syndrome"

- Remove astigmatism as this alone is not causative for Strabismics being unable to fixate the two eyes on a single target point in space.

- Deleted astigmatism and instead included extra-ocular muscle paralysis.

Page 7, line 3 – 4:

Authors have added the following:

"Amblyopia onset is most often associated with strabismus in early childhood and intermitted strabismus is often observed among amblyopes in adulthood."

- Do author mean to say the strabismus persists into adulthood in amblyopia? The last part of the sentence is unclear, as amblyopes could have both constant as well as intermittent strabismus in adulthood or in case of refractive amblyopia, no manifest deviation may also occur.

- Based on the comment we have changed the wording to: Amblyopia onset is most often associated with strabismus in early childhood and persists into adulthood, as adult amblyopes could have both constant as well as intermitted strabismus

- Report Mauchly's Tests of Sphericity prior to the main results.

- We have moved the Mauchly's Test of Sphericity prior to the main results.

Other Minor comments:

Add units to age (years)

- Added this in text and in the tables

Use appropriate units for stereoacuity. In the manuscript it is recorded as e.g. 800' to 40' arcsecs. The Unit for second of arc is: ". Using just one either " or arcsecs, would be adequate once introduced at start

- Addressed this by leaving "arcsecs" in the text and " in the tables.